# FAIR Universe HiggsML Uncertainty Dataset and Competition

Wahid Bhimji[1], Ragansu Chakkappai[2,3], Po-Wen Chang[1], Yuan-Tang Chou[4], Sascha Diefenbacher[1], Jordan Dudley[1,5], Ibrahim Elsharkawy[6,12], Steven Farrell[1], Aishik Ghosh[1,7,12], Cristina Giordano[8], Isabelle Guyon[2,9], Chris Harris[1], Yota Hashizume[10], Shih-Chieh Hsu[4], Elham E Khoda[1,4,11], Claudius Krause[8], Ang Li[8], Benjamin Nachman[1,14], David Rousseau[2,3], Robert Schoefbeck[8], Maryam Shooshtari[8], Dennis Schwarz[8], Ihsan Ullah[3], Daohan Wang[8], and Yulei Zhang[4]

[1]Lawrence Berkeley National Laboratory, [2]Université Paris-Saclay, CNRS/IN2P3, IJCLab, [3]ChaLearn, USA, [4]University of Washington, Seattle, [5]University of California, Berkeley, [6]University of Illinois Urbana-Champaign, [7]University of California, Irvine, [8]Institute for High Energy Physics, Vienna, [9]Université Paris-Saclay, [10]Kyoto University, [11]University of California, San Diego, [12]now at University of Toronto, [13]now at Georgia Institute of Technology, [14]now at Stanford University

## Abstract

The FAIR Universe – HiggsML Uncertainty Challenge focused on measuring the physical properties of elementary particles with imperfect simulators. Participants were required to compute and report confidence intervals for a parameter of interest regarding the Higgs boson while accounting for various systematic (epistemic) uncertainties. The dataset is a tabular dataset of 28 features and 280 million instances. Each instance represents a simulated proton-proton collision as observed at CERN's Large Hadron Collider in Geneva, Switzerland. The features of these simulations were chosen to capture key characteristics of different types of particles. These include primary attributes, such as the energy and three-dimensional momentum of the particles, as well as derived attributes, which are calculated from the primary ones using domain-specific knowledge. Additionally, a label feature designates each instance's type of proton-proton collision, distinguishing the Higgs boson events of interest from three background sources. As outlined in this paper, the permanent dataset release allows long-term benchmarking of new techniques. The leading submissions, including Contrastive Normalising Flows and Density Ratios estimation through classification, are described. Our challenge has brought together the physics and machine learning communities to advance our understanding and methodologies in handling systematic uncertainties within AI techniques.

## 1 Introduction

### 1.1 Background and impact

For several decades, the discovery space in almost all branches of science has been accelerated dramatically due to increased data collection brought on by the development of larger, faster instruments. More recently, progress has been further accelerated by the emergence of powerful AI approaches, including deep learning, to exploit this data. However, an unsolved challenge that remains, and *must* be tackled for future discovery, is how to effectively quantify and reduce uncertainties, including understanding and controlling *systematic* uncertainties (also named *epistemic* uncertainties in other fields). A compelling example is found in analyses to further our fundamental understanding of the

39th Conference on Neural Information Processing Systems (NeurIPS 2025) Track on Datasets and Benchmarks.

universe by analysing the vast volumes of particle physics data produced at CERN, in the Large Hadron Collider (LHC) [1]. Ten years ago, part of our team co-organised the Higgs Boson Machine Learning Challenge (HiggsML) [2, 3], the most popular Kaggle challenge at the time, attracting 1785 teams. This challenge has significantly heightened interest in applying Machine Learning (ML) techniques within High-Energy Physics (HEP) and, conversely, has exposed physics issues to the ML community. Whereas previously, the most effective methods predominantly relied on boosted decision trees, Deep Learning has since gained prominence (see, e.g., HEP ML living review [4]).

After the Higgs boson discovery was established in 2012, the focus of the community has shifted from discovery mode to precision physics mode, from the vast amount of data (tens of Petabytes) being collected. Measuring the Higgs boson's properties isn't just about studying an elusive particle; it's about probing the Higgs field itself, a fundamental component of the vacuum that has existed everywhere, since the beginning of time (the Big Bang).

High-energy physics relies on statistical analysis of aggregated observations. Therefore, the interest in uncertainty-aware ML methods in HEP is nearly as old as the application of ML in the field. Advanced efforts that integrate uncertainties into the ML training include approaches that explicitly depend on *Nuisance Parameters*[1] [5–14], that are insensitive to Nuisance Parameters [15–32], that use downstream test statistics in the initial training [33–43], and that use Bayesian neural networks for estimating uncertainties [44–47]. Many of these topics were covered in recent forward-looking review-type articles in Refs. [48–50]. However, these developments all report technique performance on different ad-hoc datasets, so it is difficult to compare their merits. The Fair Universe HiggsML Uncertainty Challenge, an official NeurIPS 2024 competition, aimed to provide a common ground, with a dataset of sufficient complexity, equipped with systematic bias parameterisations, and a metric.

We aim to address the issue of systematic uncertainties within a specific domain. Yet, the techniques developed by the challenge participants will apply to identifying, quantifying, and correcting systematic uncertainties in other areas, particularly other science disciplines.

## 1.2 Novelty

This entirely new public competition has been built on our experience running several competitions in particle physics and beyond. These include the original HiggsML challenge [2], the TrackML Challenges (NeurIPS 2018 competition) [51, 52], the LHC Olympics [53], AutoML/AutoDL [54, 55], and other competitions. Building on the foundation of the HiggsML challenge, this competition introduces a significant change by using simulated data that includes biases (or *systematic effects*). In addition, participants were asked to provide a confidence interval and not just a point estimate.

While there have been previous challenges focusing on meta-learning and transfer-learning, such as the NeurIPS 2021 and 2022 meta-learning challenges [56, 57], Unsupervised and Transfer Learning [58], challenges related to bias e.g. Crowd bias challenge [59], and those addressing distribution shifts, like the Shifts challenge[60] series, and CCAI@UNICT 2023 [61], this is the first challenge and dataset that requires participants to handle systematic uncertainty. Moreover, this project is connecting the Perlmutter system at NERSC [62], a large-scale supercomputing resource featuring over 7000 NVIDIA A100 GPUs, with Codabench [63], a new version of the renowned open-source benchmark platform Codalab [64, 65]. Due to its complexity, the process of generating events was computationally intensive; use of the Perlmutter supercomputer allowed us to create a vast amount of data which will serve as a long-lasting benchmark – hundreds of millions of events compared to less than a million events for the HiggsML competition.

## 2 Data

The dataset provided is tabular, where each row is a high-energy simulated proton-proton collision by the ATLAS experiment [66] at the LHC. Figure 1 represents the chosen final state. The events are divided into two categories (see Table 1): signal and background. The signal category includes collision events with a Higgs boson decaying into pairs of tau particles (one decaying into a light lepton and neutrinos, the other one into a set of hadrons and one neutrino hence the name hadronic

---

[1]The name Nuisance Parameter, commonly used in the physics literature, refers to a parameter governing a specific parameterisation of a systematic bias. Nuisance Parameters can be in part constrained from the data itself. Still, the name implies that constraining them is only interesting as an auxiliary task.

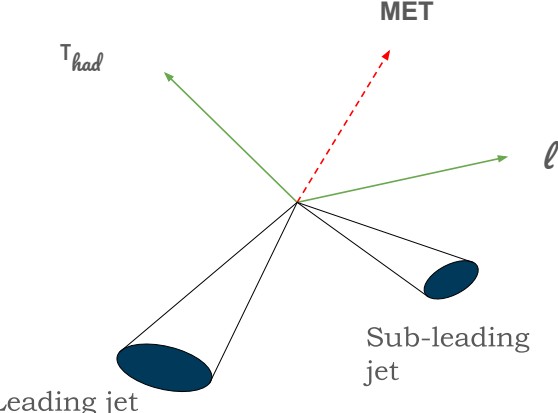

Figure 1: Diagram of the particles in the final state chosen: one lepton, one tau hadron, up to two jets, and the missing transverse momentum vector.

tau), while the background category includes other processes (subcategories) leading to a similar final state, but without an intermediate Higgs boson. Neutrinos can be measured only indirectly through the missing transverse momentum vector. The two highest transverse momentum jets can also contribute to the final state, default values are reported if no or only one jet. Of the 28 features, 16 are primary features, essentially the energy and direction of particles in the final state, and 12 are derived features computed from the primary ones with domain knowledge. Three additional features provide ground truth information, allowing for supervised learning. A much more detailed description is available in appendices A, B, and C; it is mainly taken from the public previously unpublished Fair Universe whitepaper [67], which served as detailed documentation for the competition.

The dataset was created by chaining two widely-used simulation tools, Pythia 8.2 [68] and Delphes 3.5.0 [69]; all the configuration and data pre-selection code is available from [70]. The production required 1.8 million CPU core hours; software commissioning runs only contributed in a negligible way to the resource usage.

The dataset is publicly available on the Zenodo platform [71], under license CC-BY 4.0. The data is saved as a tabular parquet [72] file of 16 GB and is accompanied by a Croissant JSON metadata file. The dataset comprises 280M simulated proton-proton collision events and is weighted to represent two weeks of LHC data taking. A separate 120M i.i.d dataset has been used for the final results in section 5 and is kept private for future over-training checks.

In addition, we provide a biasing script [73] capable of manipulating a dataset by introducing six parameterised distortions as a function of six corresponding Nuisance Parameters; see details in Appendix D. For example, a detector miscalibration can cause a bias in other features in a cascading way, or in another case, the magnitude of a particular background (e.g. the $t\bar{t}$) contribution can change so that the feature distributions can be different. In both cases, the inference would be done on a dataset not i.i.d. to the training dataset.

## 3  Tasks and application scenarios

The participant's objective is to develop an estimator for the number of Higgs boson events in a dataset analogous to results from a LHC experiment. Such a measurement is typical of those carried out at the LHC, which allows us to strengthen (or invalidate) our understanding of the fundamental laws of nature.

The parameter of interest is the *signal strength* ($\mu$), which is the number of estimated Higgs boson events divided by the number of such events predicted by the Standard Model, which is the reference theory. The challenge involved estimating $\mu$'s true value, $\mu_{true}$, which may vary from one (in practice for the challenge in the range 0.1 to 3) and is inherently unknown.

Participants were tasked with generating a 68.27% Confidence Interval (CI) for $\mu$, incorporating both aleatoric (random) and epistemic (systematic) uncertainties rather than a single-point estimate. The six different systematic uncertainties are implemented in Appendix D.

The primary simulation dataset assumes a $\mu$ of one. Participants use a training subset, where events are labelled based on their event type (Higgs boson event, or background). We provide a script to generate unlabelled *pseudo-experiment* datasets from the primary simulation dataset for any value of $\mu$ and the six systematic biases. A pseudo-experiment is a test dataset corresponding to what could be collected from running the Large Hadron Collider for 10 fb$^{-1}$, corresponding to approximately 800 billion inelastic proton collisions. The participant's model should be able to reverse the process and provide a 68.27% CI on $\mu$ for any pseudo-experiment. The task could be seen as a regression of $\mu$ and the six Nuisance Parameters, but the fact that the metric only concerns one of the regressed parameters ($\mu$) makes it special.

In a machine learning context, the task resembles a transduction problem with distribution shift: it requires constructing a $\mu$ interval estimator from labelled training data and biased unlabelled test data. One possibility is to train a classifier to distinguish the Higgs boson from the background, with robustness against bias achieved possibly through data augmentation (or an adversarial approach, or black box optimisation or any other novel approach) via the provided script.

This challenge shifts focus from the qualitative discovery of individual Higgs boson events (which was the focus of our first challenge [2]) to the quantitative estimation of overall Higgs boson counts in test sets, akin to assessing disease impact on populations rather than diagnosing individual cases.

### 3.1 Metrics

Participants provided a model that can analyse a pseudo-experiment to determine $(\mu_{16}, \mu_{84})$, the bounds of the 68.27% (one standard deviation) Confidence Interval (CI) for $\mu$. The model is evaluated from the set of $[\mu_{16,i}, \mu_{84,i}]$ intervals obtained from $N_{\text{test}}$ pseudo-experiments, see Figure 2a. The model's performance is assessed based on two criteria:

**Average Interval Width**: $w$ (the smaller the better) computed as $w = \frac{1}{N_{test}} \sum_{i=1}^{N} |\mu_{84,i} - \mu_{16,i}|$.

**Coverage**: the frequency with which $\mu_{\text{truth}}$ is covered by the CI (the closer to the standard 68.27% probability the better) computed as $c = \frac{1}{N_{test}} \sum_{i=1}^{N} \mathbb{I}_{\mu_{\text{true},i} \in [\mu_{16,i}, \mu_{84,i}]}$. A penalising function $f$ is defined to penalise the departure of $c$ from the expected 68.27%, taking into account $\sigma_{68} = \sqrt{\frac{(1-0.6827)0.6827}{N_{\text{test}}}}$ the binomial statistical error on $c$ if $c$ is 68.27% as expected:

$$f(c) = 1 + \mathbb{I}_{c<0.6827-2\sigma_{68}} \cdot \left| \frac{c - (0.6827 - 2\sigma_{68})}{\sigma_{68}} \right|^4 + \mathbb{I}_{c>0.6827+2\sigma_{68}} \cdot \left| \frac{c - (0.6827 + 2\sigma_{68})}{\sigma_{68}} \right|^3 \quad (1)$$

We opted for an asymmetric penalty function because, within the High Energy Physics (HEP) field, overestimating uncertainty is deemed more acceptable than underestimating it [74, 75]. Hence, coverage exceeding 68.27% incurs a lesser penalty than coverage falling below 68.27%.

The final **Quantile Score** (the larger the better) used to rank participants is calculated as follows:

$$\text{score} = -\ln((w + \epsilon)f(c)), \quad (2)$$

$w$ represents the average width of the Confidence Interval, $c$ is the coverage, and $\epsilon = 10^{-2}$ is a regularisation term to guard against submissions reporting unrealistically narrow CIs. To ensure efficient use of resources, each participant's model inference was executed during the competition across 100 pseudo-experiments times 10 trials, each with distinct values of $\mu_{\text{truth}}$, with a time limit of 20s per inference on CPU or GPU. In the Final phase of the competition, each participant's best submission was evaluated over 100 pseudo-experiments, times 1000 trials, to minimise the statistical variance.

### 3.2 Limitations

The main limitation of the setup is that biases can be exactly parameterised: we are in the "Known Unknowns" regime. "Unknown Unknowns", unexpected biases, are not covered.

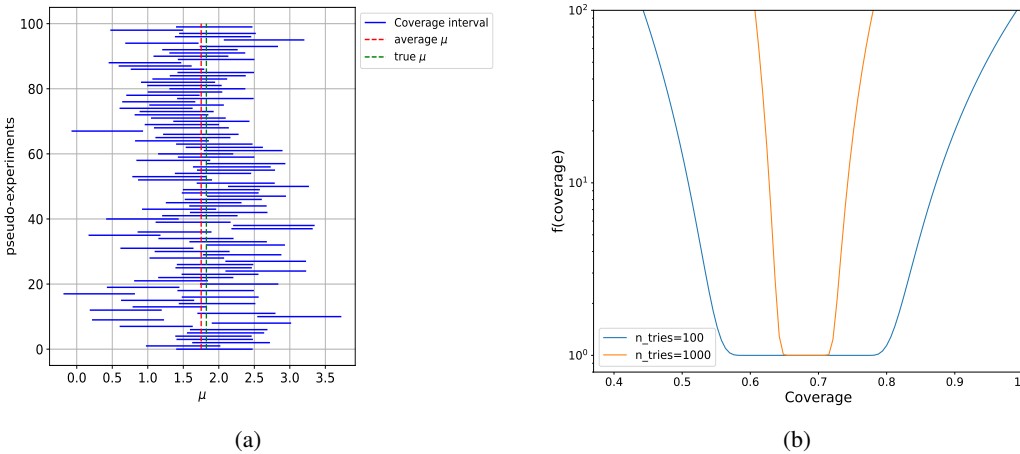

(a)

(b)

Figure 2: ((a) *Coverage plot*: all the predicted intervals (blue lines) for each pseudo experiment generated for a given $\mu_{\text{true}}$ (vertical dotted line). The coverage (here $70 \pm 5\%$) is determined by the fraction of time the vertical line intersects the horizontal blue lines. (b) Penalising function as a function of the coverage value $c$, for two values of $N_{test}$, the number of pseudo-experiments.

The dataset has been produced using well-known standard software for event generation and detector simulation. However, a proper physics measurement would require more complex software, like Madgraph [76] for more precise theoretical calculations and Geant4 [77] for detailed detector description, which are several orders of magnitude slower, yielding marginally different simulated data. The methods developed on our dataset would perform equally well, provided they are fully retrained.

The features provided for each instance of the datasets are essentially the energy and direction of a small set of particles and derived quantities. A real physics measurement may also rely on additional quantities related to the quality of particle identification or to other particles in the same proton-proton collision. Nevertheless, the algorithms developed on our dataset should require limited added complexity to deal with additional features.

## 4   Software

Alongside the dataset, a GitHub repository [73] with the relevant code for reading and analysing it is made available. This includes a Jupyter notebook starting kit, simple baseline models, a small dataset sample, and code to compute the score.

The **Starting Kit** kit includes code for installing necessary packages, loading and visualising data, training and evaluating a model with the metrics described in subsection 3.1. The **Baseline** method estimates $\mu$ using standard (for particle physics) techniques without directly addressing systematic uncertainties for simplicity. Initially, it utilises a classifier (based on an XGBoost Boosted Decision Tree) trained on a subset of the training data to build summary statistics that enhance the relative signal event density and reduce the $\mu$ estimator variance. The classifier's decision threshold is fixed heuristically. $\mu$ is then estimated from these filtered events, assuming a Poisson distribution, allowing interval maximum likelihood estimation. Further refinement involves binning events based on their classifier score and estimating $\mu$ per bin. A holdout dataset is used to build templates with the amount of background and signal in each bin for $\mu = 1$. For each pseudo-experiment, a maximum likelihood fit from the templates permits estimating $\mu$ (and the corresponding CI). On Figure 3a, the alignment of maximum likelihood estimation (orange line) with unlabelled data (black line) indicates the method's success, in the absence of any bias.

When unknown biases occur, the prediction on the amount of background and signal events per bin will be wrong, biasing the estimation of $\mu$. To address the problem of systematic errors, we use the holdout dataset with biases by different amounts of the Nuisance Parameter ($\theta$) and then build a calibration curve to estimate the signal and background in each bin. Figure 3b shows one such fit curve for the 24th bin (just as an example). Now, instead of $\mu$ depending on $S$ and $B$, it will

depend on fit functions $S(\theta)$ and $B(\theta)$. Finally, the minimisation function now regresses both $\mu$ and $\theta$, thus making the model less susceptible to systematic bias. But this is only limited to one nuisance parameter; participants are encouraged to enhance the Baseline model, for instance, by modifying the architecture or training protocol to improve resilience against biases, attempting to directly model the biases, or refining the estimator through a bias-aware model.

Another way to see it is that, armed with the biasing script which can produce a dataset for any value of the six Nuisance Parameters and the signal strength $\mu$, the participants could train a model which could regress the seven parameters for any pseudo-experiment and report the Confidence Interval on $\mu$. The winning trio did this with different techniques (section 5).

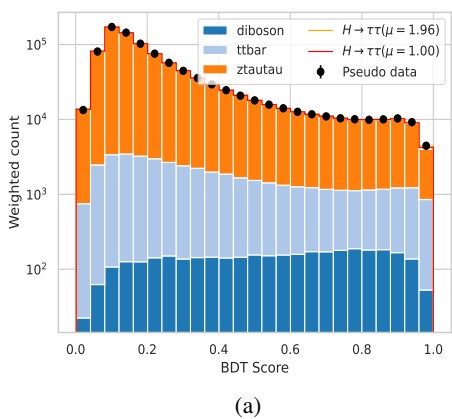
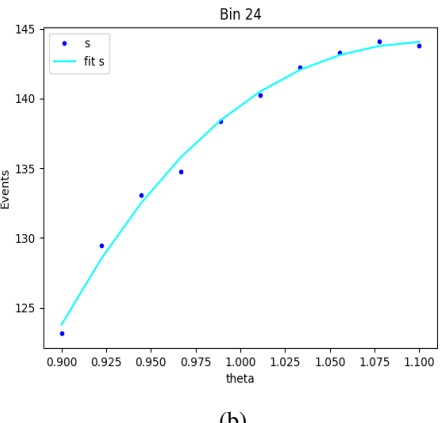

(a)                                              (b)

Figure 3: (a) classifier score for unlabelled test data (black points), and holdout data for background events $Z \to \tau\tau$ (dark orange), background $t\bar{t}$ (light blue), background di-boson (dark blue), signal events $H \to \tau\tau$ for $\mu = 1$ (red line), and signal events fitted histogram to test data, leading to estimated $\mu = 1.96$ (orange line) (b) model of the bin content vs Nuisance Parameter $\theta$ for bin 24, as an example.

## 5 Competition results and best submissions

At the end of the competition, a clear trio was at the top of the public leaderboard: HEPHY with a quantile score of 0.878, followed by Ibrahime (0.823) and Hzume (0.179). All submissions have been reevaluated on a new dataset (i.i.d. to the original one). The evaluation was done on 1000 trials of 100 pseudo-experiments (each trial with a given value of $\mu$ randomised between 0.1 and 3), instead of 10 trials for the public leaderboard. All submissions were run on the same pseudo-experiments, instead of separate pseudo-experiments for the public leaderboard.

Figure 4 shows the results for all trials for the trio. The CI width is seen falling at small and large values of $\mu$: this is due to the clipping of the Confidence Interval to a minimum value of 0.1 and a maximum value of 3 (which was not done in Figure 2a), which were the extrema values in this competition. Such clipping would be meaningless in a real physics measurement where $\mu$ is only known to be positive or null. This is the only "hack" specific to the competition context that could be identified. As far as the score is concerned, HEPHY and Ibrahim are very close. When merging all trials, the scores obtained by the top trio are: HEPHY -0.582, Ibrahim -0.576 and HZUME -2.16. An additional bootstrap analysis of the variance of these results showed that HEPHY and Ibrahim cannot be reliably ranked, hence the final rankings :

- 1st tie: team HEPHY (Lisa Benato, Cristina Giordano, Claudius Krause, Ang Li, Robert Schöfbeck, Maryam Shooshtari, Dennis Schwarz, Daohan Wang) from Vienna's Institute of High Energy Physics (HEPHY) in Austria wins $2000.
- 1st tie IBRAHIME (Ibrahim Elsharkawy) from University of Illinois at Urbana-Champaign, USA wins $2000.
- 3rd HZUME (Hashizume Yota) from Kyoto University, Japan wins $500

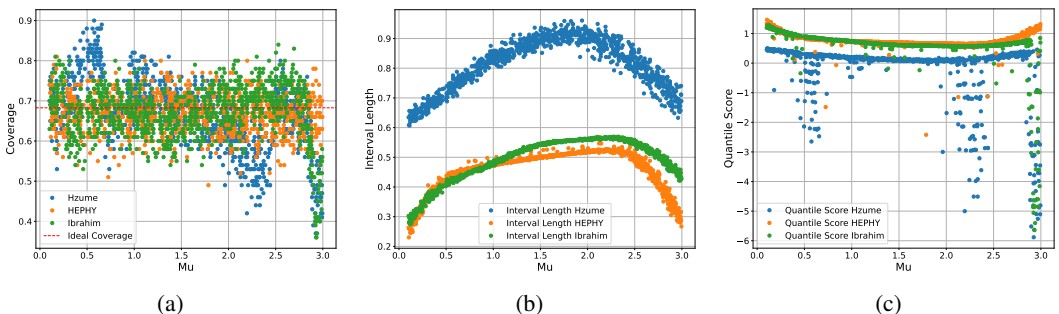

(a)             (b)             (c)

Figure 4: Comparative study of the three finalists (blue for Hzume, orange for HEPHY and green for Ibrahim's model) with 1000 trials of 100 pseudo-experiments (see subsection 3.1). 4a the coverage from each trial, 4b the average CI width and 4c the quantile score

All three are co-authors of this paper and have summarised their algorithms in the following sub-sections. HEPHY and Ibrahim's sub-sections also refer to their public full papers and code.

### 5.1   HEPHY: Simulation-based inference with a calibrated multiclassifier and parametric regressors for learning systematics

We use simulation-based inference (SBI) to construct a flexible, unbinned, and refinable surrogate of the extended likelihood [78] that captures the full high-dimensional event information for inference of the signal strength $\mu$ and the nuisance parameters $\boldsymbol{\nu}$ via a multiclassifier and parametric regressors [79]. The codebase for the "Guaranteed Optimal Likelihood-based Unbinned Method" (GOLLUM) is publicly available at Ref. [80]. We give only a brief summary here.

If we denote the likelihood by $L(\cdot)$, the integrated luminosity by $\mathcal{L}$, and the observed data set by $\mathcal{D} = \{\boldsymbol{x}_i\}_{i=1}^{N_{\mathrm{obs}}}$, the extended profile likelihood–ratio test statistic is

$$q_\mu(\mathcal{D}) = -2\log \frac{\max_{\boldsymbol{\nu}} L(\mathcal{D}|\mu,\boldsymbol{\nu})}{\max_{\mu,\boldsymbol{\nu}} L(\mathcal{D}|\mu,\boldsymbol{\nu})} = \min_{\boldsymbol{\nu}} u(\mathcal{D}|\mu,\boldsymbol{\nu}) - \min_{\mu,\boldsymbol{\nu}} u(\mathcal{D}|\mu,\boldsymbol{\nu}),$$

with

$$-\tfrac{1}{2}\,u(\mathcal{D}|\mu,\boldsymbol{\nu}) = -\mathcal{L}\big[\sigma(\mu,\boldsymbol{\nu}) - \sigma(1,\mathbf{0})\big] + \sum_{i=1}^{N_{\mathrm{obs}}} \log\left(\frac{\mathrm{d}\sigma(\boldsymbol{x}_i|\mu,\boldsymbol{\nu})}{\mathrm{d}\sigma(\boldsymbol{x}_i|1,\mathbf{0})}\right).$$

We parametrize the inclusive yield $\mathcal{L}\,\sigma(\mu,\boldsymbol{\nu})$ (total expected events) and the differential cross-section ratio $\frac{\mathrm{d}\sigma(\boldsymbol{x}|\mu,\boldsymbol{\nu})}{\mathrm{d}\sigma(\boldsymbol{x}|1,\mathbf{0})}$ (density ratios) with surrogates, trained separately in six disjoint selection regions, two of which are signal-enriched and the remainder mainly serve to constrain the nuisance parameters.

A multiclass classifier is trained on nominal (i.e., unvaried) simulated data and predicts the class probabilities for the four processes $H \to \tau\tau$, $Z \to \tau\tau$, $t\bar{t}$, and $VV$ (denoted $\hat{g}_p(\boldsymbol{x})$ for process $p$). In the likelihood, these probabilities are scaled by normalization factors $(1+\alpha)^\nu$ for the nuisance parameters $\nu_{\mathrm{bkg}}$, $\nu_{t\bar{t}}$, and $\nu_{VV}$ that control the background normalizations. The corresponding pre-fit scales $\alpha_{\mathrm{bkg}}$, $\alpha_{t\bar{t}}$, and $\alpha_{VV}$ set the sizes of these uncertainties (defined in Ref. [79]). A critical step is a dedicated, high-precision iterative isotonic regression to calibrate the classifier outputs.

To account for the dependence of the likelihood on the remaining systematic uncertainties, a second set of networks estimates the relative variation of the differential cross-section as a function of the calibration-type nuisances $\boldsymbol{\nu}_{\mathrm{calib}} = \{\nu_{\mathrm{tes}}, \nu_{\mathrm{jes}}, \nu_{\mathrm{met}}\}$. These nuisances control detector calibration uncertainties and enter the training samples via a biasing script. For each process $p$ and each selection region $r$, we fit an exponential ansatz parameterized by a neural network,

$$\frac{\mathrm{d}\sigma_p(\boldsymbol{x}|\mu,\boldsymbol{\nu})}{\mathrm{d}\sigma_p(\boldsymbol{x}|1,\mathbf{0})} \;\simeq\; \hat{S}_p(\boldsymbol{x}|\boldsymbol{\nu}_{\mathrm{calib}}) \;=\; \exp\!\big(\nu_A\,\hat{\Delta}_{p,A}(\boldsymbol{x})\big),$$

where $\nu_A$ is a multi-index enumerating the three linear, three quadratic, and three mixed terms in $(\nu_{\mathrm{tes}}, \nu_{\mathrm{jes}}, \nu_{\mathrm{met}})$, and the functions $\hat{\Delta}_{p,A}(\boldsymbol{x})$ are learned by the network.

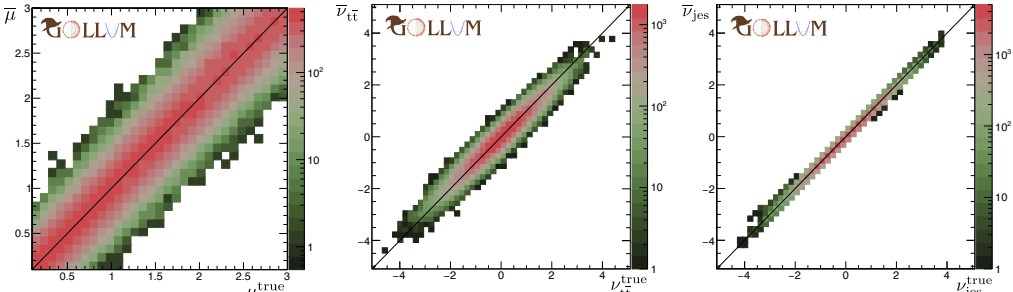

Figure 5: Scatter plot of the true value of the $H \to \tau\tau$ signal strength parameter $\mu$ (left) and the MLE $\bar{\mu}$ for $5 \cdot 10^4$ toys showing stability over the whole range of relevant $\mu_{\text{true}}$. The normalisation-type nuisance parameter $\nu_{t\bar{t}}$ (middle) and the calibration-type nuisance parameter $\nu_{\text{jes}}$ (right) are severely constrained, reducing the impact of the corresponding uncertainties.

Based on the cross-entropy loss, this ansatz leads to

$$L[\hat{\Delta}_A] = \sum_{\boldsymbol{\nu} \in \mathcal{V}} \left[ \int d\sigma(\boldsymbol{x}|\boldsymbol{0}) \, \text{Soft}^+(\nu_A \hat{\Delta}_A(\boldsymbol{x})) + \int d\sigma(\boldsymbol{x}|\boldsymbol{\nu}) \, \text{Soft}^+(-\nu_A \hat{\Delta}_A(\boldsymbol{x})) \right], \quad (3)$$

where $\mathcal{V}$ denotes the set of nuisance settings used during training and $\text{Soft}^+(x) \equiv \log(1 + e^x)$.

Thus, the surrogate can interpolate continuously in both feature and nuisance-parameter space. The complete likelihood then follows from the surrogate for the differential cross-section ratio,

$$\frac{d\sigma(\boldsymbol{x}|\mu, \boldsymbol{\nu})}{d\sigma(\boldsymbol{x}|1, \boldsymbol{0})} \simeq \mu \, \hat{g}_H(\boldsymbol{x}) \, \hat{S}_H(\boldsymbol{x}|\boldsymbol{\nu}_{\text{calib}}) + (1 + \alpha_{\text{bkg}})^{\nu_{\text{bkg}}} \Big( \hat{g}_Z(\boldsymbol{x}) \, \hat{S}_Z(\boldsymbol{x}|\boldsymbol{\nu}_{\text{calib}})$$

$$+ (1 + \alpha_{t\bar{t}})^{\nu_{t\bar{t}}} \, \hat{g}_{t\bar{t}}(\boldsymbol{x}) \, \hat{S}_{t\bar{t}}(\boldsymbol{x}|\boldsymbol{\nu}_{\text{calib}}) + (1 + \alpha_{\text{VV}})^{\nu_{\text{VV}}} \, \hat{g}_{\text{VV}}(\boldsymbol{x}) \, \hat{S}_{\text{VV}}(\boldsymbol{x}|\boldsymbol{\nu}_{\text{calib}}) \Big),$$

$$(4)$$

where $\hat{g}_p(\boldsymbol{x})$ are the calibrated outputs of the multiclassifier. The surrogate is efficient to evaluate and differentiable with respect to all parameters. For the inclusive cross-section component of the extended likelihood, we employ a spline-based interpolation scheme that reduces numerical instabilities and speeds up the evaluation during profiling.

We train one multiclass classifier and one systematics network per selection region. Closure tests show that the surrogates reproduce the shapes and normalizations of the simulated distributions across many kinematic observables and over several orders of magnitude. The unbinned surrogate is modular and refinable: new systematics or background processes can be added without retraining the entire model, mirroring standard HEP analysis workflows. This "refinable" modeling is crucial for scalability to real LHC analyses, where hundreds of nuisance parameters are typical.

We profile the nuisance parameters with MINUIT [81] and determine the 68% confidence interval (CI) by evaluating the profile likelihood as a function of $\mu$. The gain from the unbinned approach becomes evident at inference time: the surrogate improves the expected $1\sigma$ CI on the signal strength by about 20% relative to a traditional binned analysis using classifier-based templates. It also yields significantly stronger constraints on nuisance parameters – especially calibration-related ones such as $\nu_{\text{tes}}$ and $\nu_{\text{jes}}$ – reducing their impact on $\mu$ by up to 65% compared to the binned case [79].

We assess performance with $5 \cdot 10^4$ toys in Figure 5. The signal strength $\mu$ is reconstructed stably over the full range of relevant $\mu_{\text{true}}$, and the profiling strongly constrains $\nu_{t\bar{t}}$ and $\nu_{\text{jes}}$, reducing the impact of the corresponding uncertainties. The total training time was $\sim 200$ CPU core-hours.

### 5.2 ibrahime: Contrastive Normalizing Flows for Uncertainty-Aware Parameter Estimation

The full description of the method can be found in the method paper [82]. The code used to train and evaluate the method is available at [83]. A binary classifier can, in principle, estimate any model

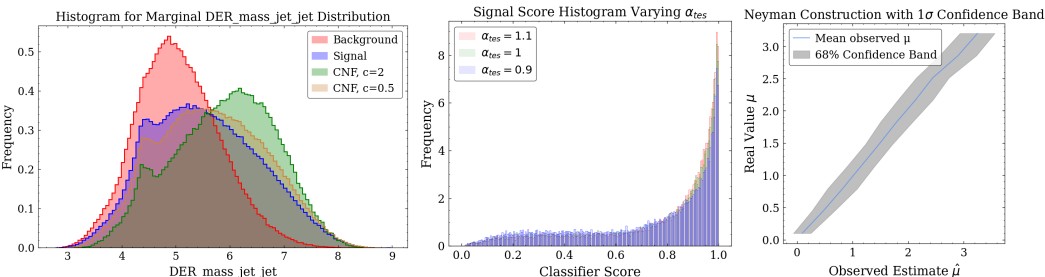

Figure 6: CNF distributions for various $c$ (left), DNN score histograms for signal varying the nuisance parameter $\alpha_{\text{tes}}$ (center panel), the Neyman confidence belt (right)

parameter $\Theta_i$ by learning a monotonic approximation of the likelihood ratio [5],

$$r(\mathbf{x}, \{\Theta_i, \nu_i\}, \{\Theta'_i, \nu'_i\}) \propto \frac{P(\mathbf{x}\,|\,\{\Theta_i, \nu_i\})}{P(\mathbf{x}\,|\,\{\Theta'_i, \nu'_i\})}, \tag{5}$$

where $\mathbf{x}$ are the data features and $\nu_i$ are nuisance parameters. In practice, this classifier approach can be impractical; if the number of model parameters $k_\Theta$ or nuisance parameters $k_\nu$ is large, the dimensionality prevents sufficient sampling of parameter space for many choices of $\{\Theta_i, \nu_i\}$ for adequate interpolation. For the challenge, $\Theta \equiv \mu \propto f_s$, where $f_s$ is the signal fraction, and $\nu_i$ are the six HiggsML nuisance parameters. Given $\mu \propto f_s$ we can attempt to learn instead the likelihood ratio $r(\mathbf{x}, \{\nu_i\}, \{\nu'_i\}) \propto \frac{p_s(\mathbf{x}\,|\,\{\nu_i\})}{p_b(\mathbf{x}\,|\,\{\nu'_i\})}$, where $p_s$ and $p_b$ are the signal and background distributions, by training on class labels and then determining $\mu$ with maximum likelihood estimation. To remedy the curse of dimensionality, we then replace the raw nuisance parameters $\nu_i$ with some discrimination functions $\Phi_{s,b}[\mathbf{x}; \{\nu_i\}]$ such that

$$r(\mathbf{x}, \{\nu_i\}, \{\nu'_i\}) \propto \frac{p_s(\mathbf{x}|\Phi_s[\mathbf{x}; \{\nu_i\}])}{p_b(\mathbf{x}|\Phi_b[\mathbf{x}; \{\nu'_i\}])}. \tag{6}$$

If these discrimination functions are relatively insensitive to nuisance parameters and take very different values for $\mathbf{x} \sim p_s$ compared to $\mathbf{x} \sim p_b$, a classifier trained on these features will more accurately approximate the desired likelihood with less data. We argue that Contrastive Normalising Flows (CNFs) are especially suitable for these functions $\Phi_{s,b}[\mathbf{x}; \{\nu_i\}]$.

**Contrastive Normalising Flows (CNFs)** A CNF is a normalising flow trained with a contrastive objective that simultaneously *maximises* the likelihood of one class and *suppresses* the likelihood of the other. Starting from the standard NF loss, and training on labelled data $\mathbf{x}_s \sim p_s$ and $\mathbf{x}_b \sim p_b$, we insert a term $c \log p_\theta^{(s)}(\mathbf{x}_b)$ so that

$$\mathcal{L}_s = \frac{1}{|\mathcal{D}|} \sum_{\mathbf{x}_s, \mathbf{x}_b \in \mathcal{D}} \left\{ -\log p_\theta^{(s)}(\mathbf{x}_s) + c \log p_\theta^{(s)}(\mathbf{x}_b) \right\} \tag{7}$$

thereby causing the learned density $p_\theta^{(s)}$ to concentrate probability mass in regions characteristic of the signal and unlike background. CNFs have been used in anomaly detection settings [84]. We generalize with $c$ and develop a novel architecture and training procedure empirically required for accurate learning [82]. Exchanging the roles of $\mathbf{x}_s$ and $\mathbf{x}_b$ gives a loss function $\mathcal{L}_b$ and a learned function $p_\theta^{(b)}$ that concentrates in background regions. Transforming these probabilities as $\Phi_{s,b}(\mathbf{x}) = p_\theta^{(s,b)}(\mathbf{x})/\left[1 + p_\theta^{(s,b)}(\mathbf{x})\right]$ gives us our monotonic discrimination functions that retain the full shape of each class. Because the model learns a class distribution, not just a decision boundary, its scores are more stable under systematic shifts than those of a purely discriminative network. Tuning $c$ lets us trade off coverage versus stability under systematic shifts seen in Figure 6.

This method can be summarized in the following steps, and required a training time of 10 GPU hours. *Step 1. Pre-processing.* Events are split into 1-jet and 2-jet categories (empirically, 0-jet events hurt performance). We take the log of features which peak near zero and then standardise all features. *Step 2. CNF density learning.* For each jet category we fit two CNFs $\left(p_{\theta,c}^{(s)}, p_{\theta,c}^{(b)}\right)$ for $c \in \{0.5, 2.0\}$. $c > 1$

sharpens signal-rich regions and is empirically shift-robust, while $c < 1$ preserves coverage. *Step 3. DNN Classifier* For any event $\mathbf{x}$ we compute $\Phi^{(s,b)}(\mathbf{x}) = \frac{p_{\theta,c}^{(s,b)}(\mathbf{x})}{1+p_{\theta,c}^{(s,b)}(\mathbf{x})}$ for $c \in \{0.5, 2.0\}$ yielding four CNF scores per jet category. Together with the primary and derived features, these are fed to a two-headed DNN (shared trunk, jet-specific heads) whose binary-cross-entropy loss is minimised on just 1,000 shifted mixtures uniformly sampling each $\nu_i$. We highlight the efficacy of CNF features with the relative invariance of the score histogram in Figure 6.

*Step 4. Maximum likelihood estimation and the Neyman Construction.* After training, the classifier scores are histogrammed for a given test set, and maximum likelihood estimation is performed to find point estimates for $\mu$, $\alpha_{jes}$, and $\alpha_{tes}$ given spline-interpolated signal and background template histograms. The point estimate for $\mu$, $\hat{\mu}$, is used to build a Neyman confidence belt, where for each value of real $\mu$ we estimate $\hat{\mu}$ and compute the 68% spread as can be seen in Figure 6. This confidence belt can then be inverted at evaluation time to find the $1\sigma$ error bars on $\mu$ given a $\hat{\mu}$ value.

### 5.3 `hzume`: Decision-Tree Aggregated Features and Hybrid Bin-Classifier/Quantile-Regressor

We build a two-stage model composed of an Aggregation stage and an Estimation stage. Total training time is one CPU core hour.

**Aggregation Stage: Feature Engineering** : (i) For each event $(x_{ij})$ a decision tree estimates the class label $y_{ij}$ (signal vs. background), yielding a probability $p_{ij}$. From the set $\{p_{ij}\}$ we compute and aggregate mean, variance, skewness, kurtosis, and the empirical quantiles at levels 0–255. (ii) For each feature $x_{ij}$ its mean and variance across events, is fed into a second decision tree that predicts the Nuisance Parameters (e.g. TES, JES). These predictions are appended as additional features.

**Estimation Stage: Two Models & Merging Strategy** (i) A decision tree classifier partitions the interval $[0.1, 3]$ into five equal-width bins and predicts the bin containing $\mu$. The resulting probability is converted into the narrowest CI covering $68\%$ of the total probability. (ii) A quantile-regression model directly predicts the lower and upper quantiles, providing an alternative CI for $\mu$.

**Model Selection Rule.** Empirically, the quantile regressor loses accuracy when $\mu$ is near the end-points (0.1 or 3). Therefore, we adopt the bin classifier in the edge regions and the quantile regressor in the central region to produce the final CI.

## 6 Conclusions and Outlook

We have prepared a dataset [71] (with relevant software [73]), challenge, and platform for developing and comparing machine learning methods that quantify uncertainties in addition to providing point estimates. With the growing size of datasets in high-energy physics, the sophistication of tools, and the precision requirements to explore new phenomena, uncertainty quantification will be an essential part of machine learning in the future. The two winning approaches, Sec. 5.1 [79] and Sec. 5.2 [82], show two alternative techniques on how the treatment of systematic uncertainties can be incorporated successfully in experimental analyses.

The two techniques have very similar performances; however, their results are not very correlated, which implies the optimum has not been reached yet. Beyond this specific metric, we expect that this unique large dataset equipped with a biasing script will be the basis of future studies, for example: (i) the precise parametrisation of density and density ratios over several order of magnitudes which is fundamental to precision physics (ii) development of morphing/Optimal Transport techniques to parameterise multidimensional non-parametric biases (iii) the same studies but with a focus on learning with a limited number of instances. Also, more complex biases could easily be introduced in the biasing script, for example, a nonlinear bias of the energy measurement, or distortions of the background contributions (instead of a scaling).

## Acknowledgements

We are grateful to the US Department of Energy, Office of High Energy Physics, and the subprogram on Computational High Energy Physics, for sponsoring this research, as well as to the ANR Chair of Artificial Intelligence HUMANIA (ANR-19-CHIA-0022). Seminal discussions contributing to this work took place at the workshop "Artificial Intelligence and the Uncertainty Challenge in Fundamental Physics," sponsored by the CNRS AISSAI Centre and the DATAIA Institute, and hosted at Institut Pascal at Université Paris-Saclay. The DATAIA Institute and Institut Pascal are respectively funded by the "Investissements d'Avenir" programs ANR-17-CONV-003 and ANR-11-IDEX-0003-01. This research used resources of the National Energy Research Scientific Computing Center (NERSC), a Department of Energy Office of Science User Facility using NERSC award HEP-ERCAP0032917. The computational results of subsection 5.1 [79] were obtained using the CLIP cluster. subsection 5.2 results were obtained with TAMU FASTER cluster at Texas A&M University through allocation 240449 from the Advanced Cyberinfrastructure Coordination Ecosystem: Services & Support (ACCESS) program [85], which National Science Foundation supports grants #2138259, #2138286, #2138307, #2137603, and #213829.

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

# A  Proton collisions and detection

This appendix gives details on how the data was generated.

The LHC collides bunches of protons every 25 nanoseconds within its four experiments. Two colliding protons produce a small firework in which part of the kinetic energy of the protons is converted into new particles. Most resulting particles are very unstable and decay quickly into a cascade of lighter particles. The ATLAS detector measures properties of these surviving particles (the so-called final state): the type of the particle (electron, photon, muon, etc.), its energy, and the 3D direction of the particle. Based on these properties, the decayed parent particle's properties can be inferred, and the inference chain continues until the heaviest primary particles are reached.

An online trigger system discards most of the bunch collisions containing uninteresting events. The trigger is a three-stage cascade classifier which decreases the event rate from 40 000 000 to about 400 per second. The selected 400 events are saved on disk, producing about one billion events and three petabytes of raw data per year.

The different types of particles or pseudo-particles of interest for the challenge are electrons, muons, hadronic tau, jets, and missing transverse energy. Electrons, muons, and taus are the three leptons[2] from the standard model.

Electrons and muons live long enough to reach the detector, so their properties (energy and direction) can be measured directly. Conversely, Taus decay almost immediately after their creation into either an electron and two neutrinos, a muon and two neutrinos, or a bunch of hadrons (charged particles) and a neutrino. The bunch of hadrons can be identified as a pseudo-particle called the hadronic tau. Jets are pseudo particles rather than real particles; they originate from a high-energy quark or gluon and appear in the detector as a collimated energy deposit associated with charged tracks. The primary information provided for the challenge is the measured momenta (see Appendix B for a short introduction to special relativity) of all the particles of the event.

We are using the conventional 3D direct reference frame of ATLAS throughout the document (see Figure 7): the $z$ axis points along the horizontal beam line, and the $x$ and $y$ axes are in the transverse plane with the $y$ axis pointing towards the top of the detector. $\theta$ is the polar angle and $\phi$ is the azimuthal angle. Transverse quantities are quantities projected on the $x - y$ plane, or, equivalently, quantities for which the $z$ component is omitted. Instead of the polar angle $\theta$, we often use the *pseudorapidity* $\eta = -\ln \tan(\theta/2)$; $\eta = 0$ corresponds to a particle in the $x - y$ plane ($\theta = \pi/2$), $\eta = +\infty$ corresponds to a particle traveling along the $z$-axis ($\theta = 0$) direction and $\eta = -\infty$ to the opposite direction ($\theta = \pi$). Particles can be identified in the $\eta$ range in $[-2.5, 2.5]$. For $|\eta| \in [2.5, 5]$, their momentum is still measured but they cannot be identified. Particles with $|\eta|$ beyond 5 escape detection along the beam pipe.

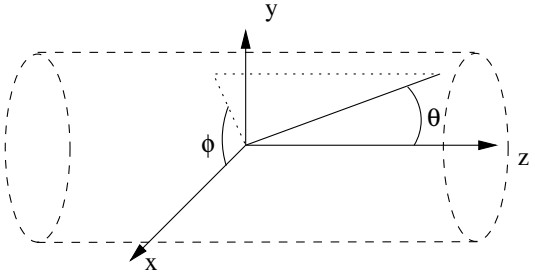

Figure 7: ATLAS reference frame

The missing transverse energy is a pseudo-particle which deserves a more detailed explanation. The neutrinos produced in the decay of a tau escape detection entirely. We can nevertheless infer their properties using the law of momentum conservation by computing the vectorial sum of the momenta of all the measured particles and subtracting it from the zero vector. In practice, measurement errors for all particles make the sum poorly estimated. Another difficulty is that many particles are lost

---

[2]For the list of elementary particles and their families, we refer the reader to http://www.sciencemag.org/content/338/6114/1558.full.

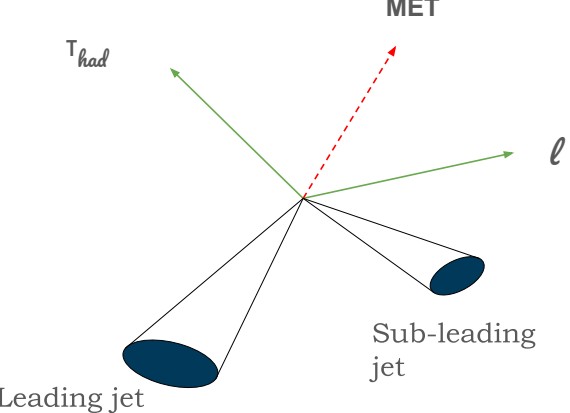

Figure 8: Diagram of the particles in the final state chosen: one lepton, one tau hadron, up to two jets, and the missing transverse momentum vector, see text for details.

Table 1: Summary of the dataset for each category and subcategory. "Number Generated" is the number of events available in the dataset. In contrast, "LHC events" is the average number in this category in a pseudo-experiment corresponding to running of the Large Hadron Collider for 10 fb$^{-1}$, corresponding to approximately 800 billion inelastic proton collisions, or 2 weeks in summer 2024 conditions.

| Process | Number Generated | LHC Events | Label |
|---|---|---|---|
| Higgs | 52 040 227 | 1 015 | **signal** |
| Z Boson | 160 383 358 | 1 002 395 | **background** |
| Di-Boson | 605 118 | 3 783 | **background** |
| $t\bar{t}$ | 7 070 398 | 44 192 | **background** |

in the beam pipe along the $z$ axis, so the information on momentum balance is lost in the direction of the $z$ axis. Thus, we can carry out the summation only in the transverse plane, hence the name missing transverse energy, which is a 2D vector in the transverse plane.

For this competition, we selected only events with exactly one electron or exactly one muon, and with exactly one hadronic tau. These two particles should be of opposite electric charge. Figure 8 shows the particles in the selected final state, whose parameters are provided in the data.

To summarise, for each event, we produce a list of momenta for an electron or muon, a tau hadron, up to two jets, plus the missing transverse energy.

Table 1 details the number of events of each category in the dataset.

# B Special relativity

This appendix gives a very minimal introduction to special relativity for a a better understanding of how the Higgs boson search is performed and what the extracted features mean (taken mainly from [86]).

## B.1 Momentum, mass, and energy

A fundamental equation of special relativity defines the so-called 4-momentum of a particle,

$$E^2 = p^2 c^2 + m^2 c^4, \tag{8}$$

where $E$ is the energy of the particle, $p$ is its momentum, $m$ is the rest mass and $c$ is the speed of light. When the particle is at rest, its momentum is zero, and so Einstein's well-known equivalence between mass and energy, $E = mc^2$, applies. In particle physics, we usually use the following units: GeV for energy, $\text{GeV}/c$ for momentum, and $\text{GeV}/c^2$ for mass. 1 GeV ($10^9$ electron-Volt) is one billion times the energy acquired by an electron accelerated by a field of 1 V over 1 m, and it is also approximately the energy corresponding to the mass of a proton (more precisely, the mass of the proton is about $1 \, \text{GeV}/c^2$). When these units are used, Equation 8 simplifies to

$$E^2 = p^2 + m^2. \tag{9}$$

To avoid the clutter of writing $\text{GeV}/c$ for momentum and $\text{GeV}/c^2$ for mass, a shorthand of using GeV for all the three quantities of energy, momentum, and mass is usually adopted in most of the recent particle physics literature (including papers published by the ATLAS and the CMS experiments). We also adopt this convention throughout this document.

The momentum is related to the speed $v$ of the particle. For a particle with non-zero mass, and when the speed of the particle is much smaller than the speed of light $c$, the momentum boils down to the classical formula $p = mv$. In special relativity, when the speed of the particle is comparable to $c$, we have $p = \gamma mv$, where

$$\gamma = \frac{1}{\sqrt{1 - (v/c)^2}}.$$

The relation holds both for the norms $v$ and $p$ and for the three dimensional vectors $\vec{v}$ and $\vec{p}$, that is, $\vec{p} = \gamma m \vec{v}$, where, by convention, $p = |\vec{p}|$ and $v = |\vec{v}|$. The factor $\gamma$ diverges to infinity when $v$ is close to $c$, and the speed of light cannot be reached or surpassed. Hence, momentum is a concept more frequently used than speed in particle physics. The kinematics of a particle is fully defined by the momentum and energy, more precisely, by the 4-momentum $(p_x, p_y, p_z, E)$. When a particle is identified, it has a well-defined mass[3], so its energy can be computed from the momentum and mass using Equation 8. Conversely, the mass of a particle with known momentum and energy can be obtained from

$$m = \sqrt{E^2 - p^2}. \tag{10}$$

Instead of specifying the momentum coordinate $(p_x, p_y, p_z)$, the parameters $\phi$, $\eta$, and $p_\text{T} = \sqrt{p_x^2 + p_y^2}$, explained in Appendix A are often used.

## B.2 Invariant mass

The mass of a particle is an intrinsic property of a particle. So, for all events with a Higgs boson, the Higgs boson will have the same mass. To measure the mass of the Higgs boson, we need the 4-momentum $(p_x, p_y, p_z, E) = (\vec{p}, E)$ of its decay products. Take the simple case of the Higgs boson $H$ decaying into a final state of two particles, $A$ and $B$, which are measured in the detector. By conservation of energy and momentum (which are fundamental laws of nature), we can write $E_H = E_A + E_B$ and $\vec{p}_H = \vec{p}_A + \vec{p}_B$. Since the energies and momenta of $A$ and $B$ are measured in the detector, we can compute $E_H$ and $p_H = |\vec{p}_H|$ and calculate $m_H = \sqrt{E_H^2 - p_H^2}$. This is called the invariant mass because (with a perfect detector) $m_H$ remains the same even if $E_H$ and $p_H$ differ from event to event. This can be generalised to more than two particles in the final state and to any number of intermediate states.

---

[3]neglecting the particle width

In our case, the final state for particles originating from the Higgs boson is a lepton, a hadronic tau, and three neutrinos. The lepton and hadronic tau are measured in the detector, but for the neutrinos, all we have is the transverse missing energy, which estimates the sum of the momenta of the three neutrinos in the transverse plane. Hence, the mass of the $\tau\tau$ can not be measured; we have to resort to different estimators which are only correlated to the mass of the $\tau\tau$. For example, the visible mass (feature DER_mass_vis) which is the invariant mass of the lepton and the hadronic tau, hence deliberately ignoring the unmeasured neutrinos. The possible jets in the events are not originating from the Higgs boson itself, but can be produced in association with it.

## B.3   Other useful formulas

The following formulas are useful to compute DERived features from PRImary features (in Appendix C). For tau, lep, leading_jet, and subleading_jet, the momentum vector can be computed as

$$
\vec{p} = \begin{pmatrix} p_x \\ p_y \\ p_z \end{pmatrix} = \begin{pmatrix} p_{\mathrm{T}} \times \cos\phi \\ p_{\mathrm{T}} \times \sin\phi \\ p_{\mathrm{T}} \times \sinh\eta \end{pmatrix},
$$

where $p_{\mathrm{T}}$ is the transverse momentum, $\phi$ is the azimuth angle, $\eta$ is the pseudo rapidity, and $\sinh$ is the hyperbolic sine function. The modulus of $p$ is

$$
p_{\mathrm{T}} \times \cosh\eta, \tag{11}
$$

where $\cosh$ is the hyperbolic cosine function. The mass of these particles is neglected, so $E = p$.

The missing transverse energy $\vec{E}_{\mathrm{T}}^{\mathrm{miss}}$ is a two-dimensional vector

$$
\vec{E}_{\mathrm{T}}^{\mathrm{miss}} = \begin{pmatrix} |\vec{E}_{\mathrm{T}}^{\mathrm{miss}}| \times \cos\phi_{\mathrm{T}} \\ |\vec{E}_{\mathrm{T}}^{\mathrm{miss}}| \times \sin\phi_{\mathrm{T}} \end{pmatrix},
$$

where $\phi_{\mathrm{T}}$ is the azimuth angle of the missing transverse energy.

The invariant mass of two particles is the invariant mass of their 4-momentum sum, that is (still neglecting the mass of the two particles),

$$
m_{\mathrm{inv}}(\vec{a}, \vec{b}) = \sqrt{\left(\sqrt{a_x^2 + a_y^2 + a_z^2} + \sqrt{b_x^2 + b_y^2 + b_z^2}\right)^2 - (a_x + b_x)^2 - (a_y + b_y)^2 - (a_z + b_z)^2}. \tag{12}
$$

The transverse mass of two particles is the invariant mass of the vector sum, but this time the third component is set to zero, which means only the projection on the transverse plane is considered. That is (still neglecting the mass of the two particles),

$$
m_{\mathrm{tr}}(\vec{a}, \vec{b}) = \sqrt{\left(\sqrt{a_x^2 + a_y^2} + \sqrt{b_x^2 + b_y^2}\right)^2 - (a_x + b_x)^2 - (a_y + b_y)^2}. \tag{13}
$$

The pseudorapidity separation between two particles, $A$ and $B$, is

$$
|\eta_A - \eta_B|. \tag{14}
$$

The $R$ separation between two particles $A$ and $B$ is

$$
\sqrt{(\eta_A - \eta_B)^2 + (\phi_A - \phi_B)^2}, \tag{15}
$$

where $\phi_A - \phi_B$ is brought back to the $]-\pi, +\pi]$ range. A good intuition for the $R$ separation is that it behaves like the 3D angle in radians between the two particles.

## C  The detailed description of the features

In this section, we explain the list of features that describe the events.

Prefix-less variables `Weight`, `Label`,`DetailedLabel`, have a special role and should not be used as regular features for the model[4]:

`Weight`  The event weight $w_i$. Not to be used as a feature. Not available in the test sample.

`Label`  The event label (integer) $y_i$ 1 for signal, 0 for background . Not to be used as a feature. Not available in the test sample.

`DetailedLabel`  The event detailed label (string) "htautau" for signal (when Label==1), "ztautau", "ttbar" and "diboson" for the three background categories (when Label==0). Not to be used as a feature. Not available in the test sample. This feature is used to implement some systematic biases; see Appendix D. It could be used to train a multi-category classifier.

The variables prefixed with `PRI` (for PRImitives) are "raw" quantities about the bunch collision as measured by the detector, essentially parameters of the momenta of particles (see Figure 9, Figure 10 and Figure 11 for their distributions).

In addition:

- Features are float unless specified otherwise.
- All azimuthal $\phi$ angles are in radian in the $]-\pi, +\pi]$ range.
- Energy, mass, and momentum are all in GeV
- All other features are unitless.
- Features are indicated as "undefined" when it can happen that they are meaningless or cannot be computed; in this case, their value is $-25$, which is outside the normal range of all variables.
- The mass of particles has not been provided, as it can safely be neglected for the challenge.

`PRI_had_pt`  The transverse momentum $\sqrt{p_x^2 + p_y^2}$ of the hadronic tau.

`PRI_had_eta`  The pseudorapidity $\eta$ of the hadronic tau.

`PRI_had_phi`  The azimuth angle $\phi$ of the hadronic tau.

`PRI_lep_pt`  The transverse momentum $\sqrt{p_x^2 + p_y^2}$ of the lepton (electron or muon).

`PRI_lep_eta`  The pseudorapidity $\eta$ of the lepton.

`PRI_lep_phi`  The azimuth angle $\phi$ of the lepton.

`PRI_met`  The missing transverse energy $\vec{E}_T^{\text{miss}}$.

`PRI_met_phi`  The azimuth angle $\phi$ of the missing transverse energy vector.

`PRI_jet_num`  The number of jets.

`PRI_jet_leading_pt`  The transverse momentum $\sqrt{p_x^2 + p_y^2}$ of the leading jet, that is the jet with the largest transverse momentum (undefined if `PRI_jet_num` $= 0$).

`PRI_jet_leading_eta`  The pseudorapidity $\eta$ of the leading jet (undefined if `PRI_jet_num` $= 0$).

`PRI_jet_leading_phi`  The azimuth angle $\phi$ of the leading jet (undefined if `PRI_jet_num` $= 0$).

`PRI_jet_subleading_pt`  The transverse momentum $\sqrt{p_x^2 + p_y^2}$ of the sub leading jet, that is, the jet with the second largest transverse momentum (undefined if `PRI_jet_num` $\leq 1$).

`PRI_jet_subleading_eta`  The pseudorapidity $\eta$ of the subleading jet (undefined if `PRI_jet_num` $\leq 1$).

---

[4]In the starting kit, they are split away in separate numpy arrays while the regular features are stored in a Dataframe

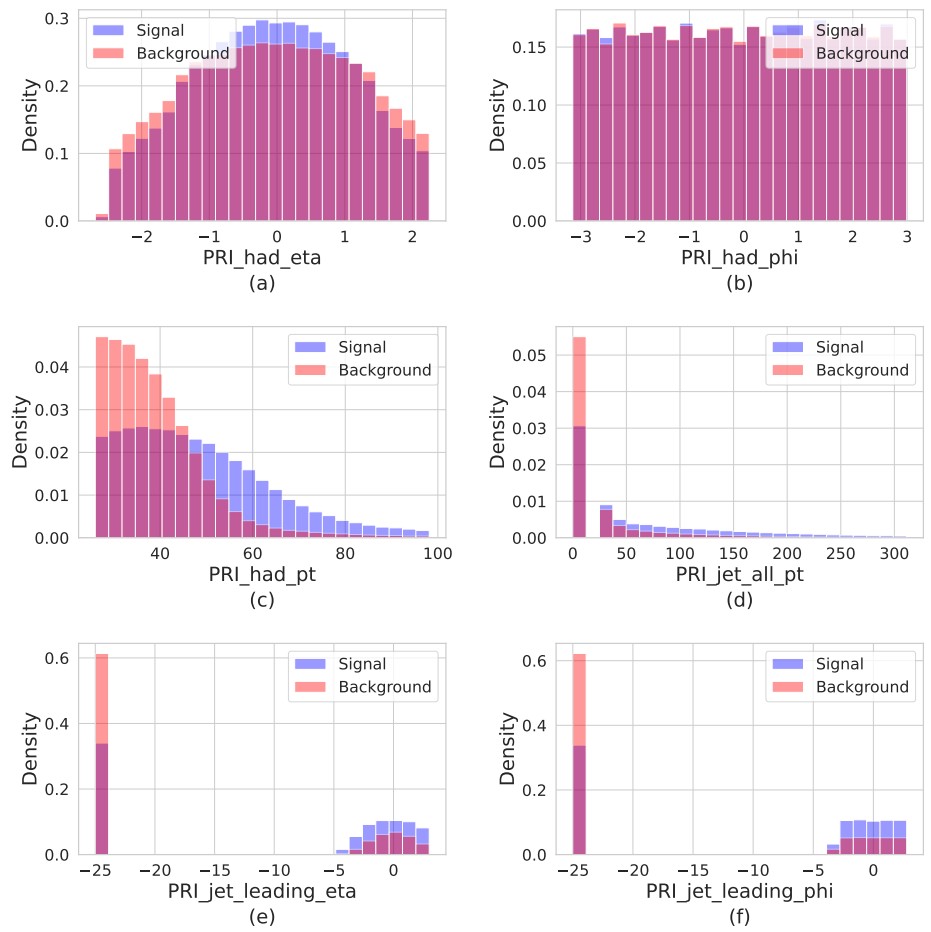

Figure 9: Distributions of: (a) hadron $\eta$, (b) hadron $\phi$, (c) hadron $p_T$, (d) all jets $p_T$, (e) leading jet $\eta$, and (f) leading jet $\phi$. For jet quantities, the left most bin is the default value in the absence of jets.

PRI_jet_subleading_phi The azimuth angle $\phi$ of the subleading jet (undefined if PRI_jet_num $\leq$ 1).

PRI_jet_all_pt The scalar sum of the transverse momentum of all the jets of the events (not limited to the first 2).

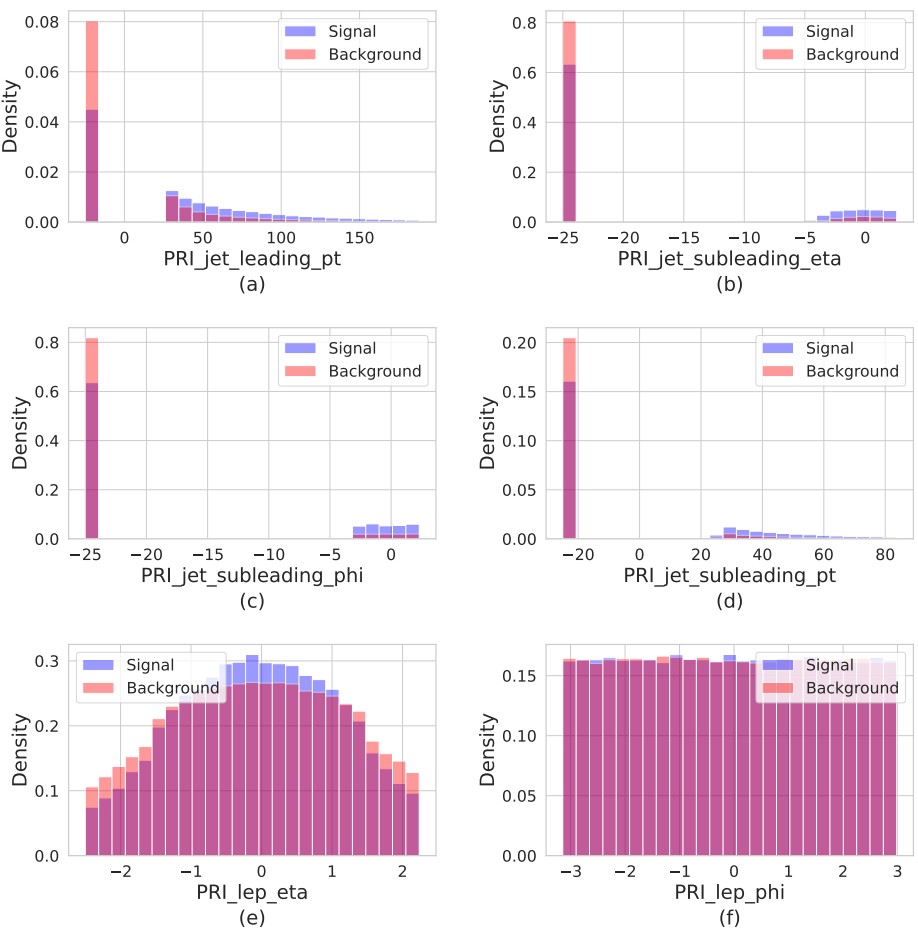

Figure 10: Distributions of: (a) leading jet $p_T$, (b) subleading jet $\eta$, (c) subleading jet $\phi$, (d) subleading jet $p_T$, (e) lepton $\eta$, and (f) lepton $\phi$. For jet quantities, the left most bin is the default value in no jet, or only one jet.

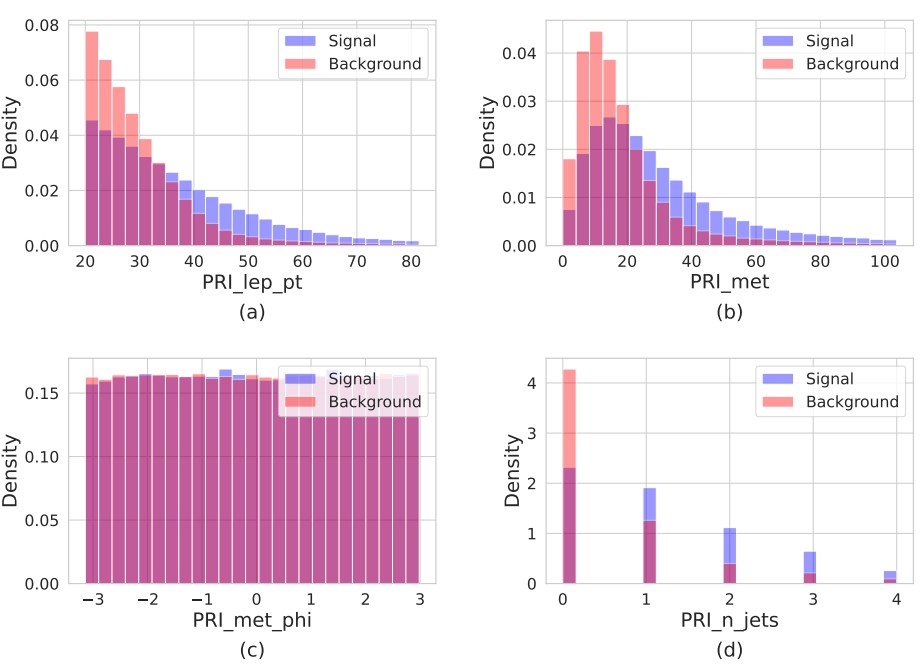

Figure 11: Distributions of: (a) lepton $p_T$, (b) MET, (c) MET $\phi$, and (d) number of jets.

Variables prefixed with DER (for DERived) are quantities computed from the primitive features on the fly from PRImary features (including possible systematics shifts )[5](see Figure 12 and Figure 13 for their distributions). These quantities were selected by the physicists of ATLAS in the reference document [87] either to select regions of interest or as features for the Boosted Decision Trees used in this analysis in order to enhance signal Higgs boson events separation from background events. DERived features were already present in the HiggsML dataset [86][6]). The DERived features correspond to feature engineering; an ideal model to be trained on infinite statistics should not need these features. This distinction between primary and derived features (or "low-level" and "high-level" or "raw variables" and "human-assisted variables") is rather standard in the AI for HEP literature, see for example [88, 89]. There is no guarantee that all DERived features are useful for this challenge (they could even be detrimental in the context of systematics). The challenge participant is free to keep these DERived features, remove them altogether, keep a few, or do more feature engineering.

DER_mass_transverse_met_lep The transverse mass (Equation 13) between the missing transverse energy and the lepton.

DER_mass_vis The invariant mass (Equation 12) of the hadronic tau and the lepton.

DER_pt_h The modulus (Equation 11) of the vector sum of the transverse momentum of the hadronic tau, the lepton, and the missing transverse energy vector.

DER_deltaeta_jet_jet The absolute value of the pseudorapidity separation (Equation 14) between the two jets (undefined if PRI_jet_num $\leq$ 1).

DER_mass_jet_jet The invariant mass (Equation 12) of the two jets (undefined if PRI_jet_num $\leq$ 1).

DER_prodeta_jet_jet The product of the pseudorapidities of the two jets (undefined if PRI_jet_num $\leq$ 1).

DER_deltar_had_lep The $R$ separation (Equation 15) between the hadronic tau and the lepton.

DER_pt_tot The modulus (Equation 11) of the vector sum of the missing transverse momenta and the transverse momenta of the hadronic tau, the lepton, the leading jet (if PRI_jet_num $\geq$ 1) and the subleading jet (if PRI_jet_num $=$ 2) (but not of any additional jets).

DER_sum_pt The sum of the moduli (Equation 11) of the transverse momenta of the hadronic tau, the lepton, the leading jet (if PRI_jet_num $\geq$ 1) and the subleading jet (if PRI_jet_num $=$ 2) and the other jets (if PRI_jet_num $>=$ 3).

DER_pt_ratio_lep_tau The ratio of the transverse momenta of the lepton and the hadronic tau.

DER_met_phi_centrality The centrality of the azimuthal angle of the missing transverse energy vector w.r.t. the hadronic tau and the lepton

$$C = \frac{A + B}{\sqrt{A^2 + B^2}},$$

where $A = \sin(\phi_{\text{met}} - \phi_{\text{lep}}) * \text{sign}(\sin(\phi_{\text{had}} - \phi_{\text{lep}}))$, $B = \sin(\phi_{\text{had}} - \phi_{\text{met}}) * \text{sign}(\sin(\phi_{\text{had}} - \phi_{\text{lep}}))$, and $\phi_{met}$, $\phi_{\text{lep}}$, and $\phi_{\text{had}}$ are the azimuthal angles of the missing transverse energy vector, the lepton, and the hadronic tau, respectively. The centrality is $\sqrt{2}$ if the missing transverse energy vector $\vec{E}_{\text{T}}^{\text{miss}}$ is on the bisector of the transverse momenta of the lepton and the hadronic tau. It decreases to 1 if $\vec{E}_{\text{T}}^{\text{miss}}$ is collinear with one of these vectors and it decreases further to $-\sqrt{2}$ when $\vec{E}_{\text{T}}^{\text{miss}}$ is exactly opposite to the bisector. The logic behind this feature is that if the neutrinos are colinear to the lepton and the hadronic tau (which is a good approximation), then the missing transverse energy vector should be between the lepton and the hadronic tau.

DER_lep_eta_centrality The centrality of the pseudorapidity of the lepton w.r.t. the two jets (undefined if PRI_jet_num $\leq$ 1)

$$\exp\left[\frac{-4}{(\eta_1 - \eta_2)^2}\left(\eta_{\text{lep}} - \frac{\eta_1 + \eta_2}{2}\right)^2\right],$$

---

[5]The code to compute DERived features from PRImitive features can be seen at https://github.com/FAIR-Universe/FAIR_Universe_dataset/blob/main/hep_challenge/derived_quantities.py

[6]The notable exception of DER_mass_MMC which was in the HiggsML dataset but is deliberately absent from the Fair-Universe dataset because it was the result of a complex and lengthy Monte-Carlo Markov Chain integration which is not practical to rerun.

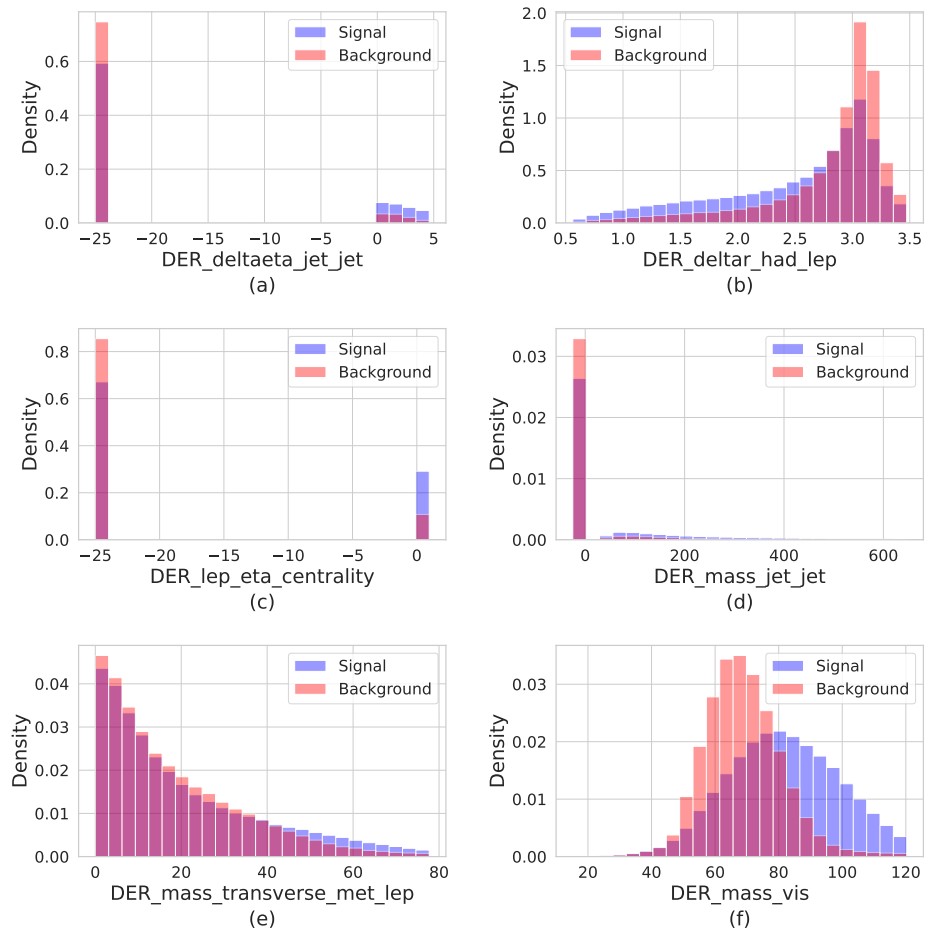

Figure 12: Distributions of kinematic variables: (a) $\Delta\eta(jet\text{-}jet)$, (b) $\Delta R(had\text{-}lep)$, (c) $lep\ \eta$ centrality, (d) $m(jet\text{-}jet)$, (e) $m_T(\text{MET-}lep)$, and (f) visible mass.

where $\eta_{\text{lep}}$ is the pseudorapidity of the lepton and $\eta_1$ and $\eta_2$ are the pseudorapidities of the two jets. The centrality is 1 when the lepton is on the bisector of the two jets, decreases to $1/e$ when it is collinear to one of the jets, and decreases further to zero at infinity. The logic behind this feature is that if the two jets are emitted together with the Higgs boson, then the Higgs decay product should be in average between the two jets.

The feature list and event sample are primarily inspired from [87]. One crucial difference is that the dataset was produced with a more straightforward (leading-order) event generator (Pythia), and the detector effect was simulated with a more straightforward detector simulation (Delphes rather than Geant4 ATLAS Simulation). These simplifications allowed us to provide to participants a large sample allowing the development of sophisticated models while preserving the complexity of the original problem.

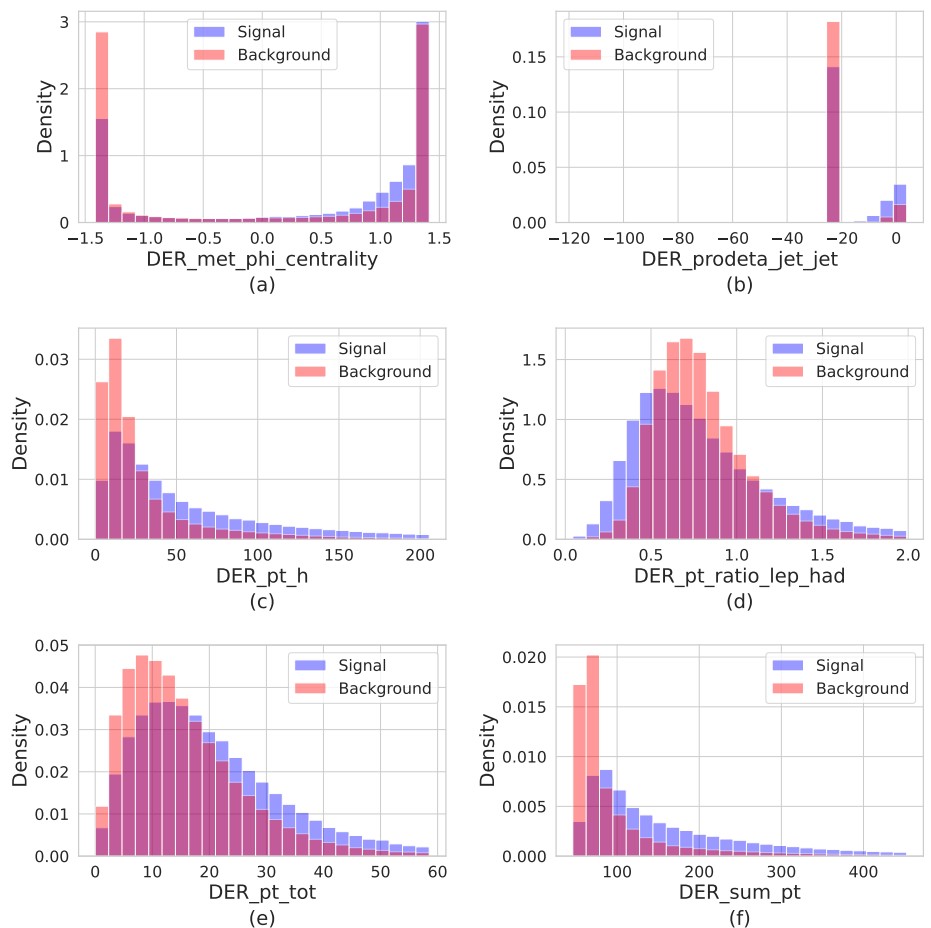

Figure 13: Distributions of: (a) MET $\phi$ centrality, (b) $\mathrm{prod}\,\eta(jet\text{-}jet)$, (c) $p_T^h$, (d) $p_T(lep/had)$ ratio, (e) $p_T^{\mathrm{tot}}$, and (f) $\sum p_T$.

| Variable | Mean | Sigma | Range |
|----------|------|-------|-------|
| $\alpha_{\text{tes}}$ | 1. | 0.01 | [0.9, 1.1] |
| $\alpha_{\text{jes}}$ | 1. | 0.01 | [0.9, 1.1] |
| $\alpha_{\text{soft\_met}}$ | 0. | 1. | [0., 5.] |
| $\alpha_{\text{ttbar\_scale}}$ | 1. | 0.02 | [0.8, 1.2] |
| $\alpha_{\text{diboson\_scale}}$ | 1. | 0.25 | [0., 2.] |
| $\alpha_{\text{bkg\_scale}}$ | 1. | 0.001 | [0.99, +1.01] |

Table 2: List of six systematic bias Nuisance Parameters defined in the challenge, with the mean and sigma of their Gaussian (Log-normal for $\alpha_{\text{soft\_met}}$) distribution and their range. The corresponding $\alpha$ is set to the Mean value whenever a systematic bias is switched off. "No systematics" means all $\alpha$ are set to their Mean value.

## D  Systematic biases

This appendix details the implementation of the systematic biases Nuisance Parameters[7].

### D.1  Systematic bias definition

Table 2 lists the different Nuisance Parameters with their Gaussian distribution and the range to which they are clipped. $\alpha_{\text{tes}}$, $\alpha_{\text{jes}}$, and $\alpha_{\text{soft\_met}}$ impacts some PRImary features, and then DERived features in cascade. $\alpha_{\text{tes}}$ and $\alpha_{\text{jes}}$ also impact which events make it to the final dataset. $\alpha_{\text{ttbar\_scale}}$, $\alpha_{\text{diboson\_scale}}$ and $\alpha_{\text{bkg\_scale}}$ only impact the Weight of some background categories, that is to say, the composition of the background (for $\alpha_{\text{ttbar\_scale}}$ and $\alpha_{\text{diboson\_scale}}$) or the overall level of the background $\alpha_{\text{bkg\_scale}}$. The Gaussian distributions parameterise our ignorance of the exact value of the biases. We think their value is 1 (or zero for $\alpha_{\text{soft\_met}}$) while their real value is slightly different, as parameterised by their width, thus biasing our measurement by an unknown amount, which can be simulated.

### D.2  Impact of biases on features

To detail the impact of the systematics, we need to detail first how the 4-momenta from the final state particles can be reconstructed from the PRImary features, following Appendix B. The four parameters ($P_x$,$P_y$,$P_z$,E) of the four-vector of each particle in the final state can be reconstructed from the PRImary features as follows (using the hadronic tau as an example, and reminding that the mass is neglected so that $E = P$),

$$P_{\text{had}} = \begin{pmatrix} \texttt{PRI\_had\_pt} * \cos(\texttt{PRI\_had\_phi}) \\ \texttt{PRI\_had\_pt} * \sin(\texttt{PRI\_had\_phi}) \\ \texttt{PRI\_had\_pt} * \sinh(\texttt{PRI\_had\_eta}) \\ \texttt{PRI\_had\_pt} * \cosh(\texttt{PRI\_had\_eta}) \end{pmatrix}$$

(where sinh and cosh are the hyperbolic sine and cosine functions), and similarly for $P_{\text{lep}}$, $P_{\text{leading jet}}$ and $P_{\text{subleading jet}}$.

The Missing ET vector is, by definition, in the transverse plane, so we have:

$$P_{\text{MET}} = \begin{pmatrix} \texttt{PRI\_met} * \cos(\texttt{PRI\_met\_phi}) \\ \texttt{PRI\_met} * \sin(\texttt{PRI\_met\_phi}) \\ \texttt{PRI\_met} \end{pmatrix}$$

$\alpha_{\text{tes}}$ is meant to describe the fact that the detector is not calibrated correctly for the measurement of the hadron momentum, meaning when the detector reports a momentum $P_{\text{had}}$ it really is :

$$P_{\text{had}}^{\text{biased}} = \alpha_{\text{tes}} P_{\text{had}}$$

And similarly, for the jets momentum (when they are defined)

$$P_{\text{jet\_leading}}^{\text{biased}} = \alpha_{\text{jes}} P_{\text{jet\_leading}}$$

---

[7]See also `https://github.com/FAIR-Universe/FAIR_Universe_dataset/blob/main/hep_challenge/systematics.py`

$$P_{\text{jet\_subleading}}^{\text{biased}} = \alpha_{\text{jes}} P_{\text{jet\_subleading}}$$

$\alpha_{\text{tes}}$ and $\alpha_{\text{jes}}$ also have an impact on $P_{\text{MET}}$: $P_{\text{MET}}$ is obtained from the opposite of the sum of all visible objects in the event so that changing one of the visible objects (like $P_{\text{had}}$, $P_{\text{leading jet}}$ or $P_{\text{subleading jet}}$) has a correlated impact on $P_{\text{MET}}$ (this calculation is performed on the first two coordinates and $E_{\text{MET}}$ is recalculated from their modulus):

$$P_{\text{MET}}^{biased} = P_{\text{MET}} + (1 - \alpha_{\text{tes}})P_{\text{had}} + (1 - \alpha_{\text{jes}})P_{\text{leading jet}} + (1 - \alpha_{\text{jes}})P_{\text{subleading jet}}$$

$\alpha_{\text{soft\_met}}$ has a different role; it expresses an additional noise source in the measurement of the missing ET vector, which is not present in the simulation. A random 2D vector of norm $ET_{\text{soft}} = \text{Lognormal}(\alpha_{\text{soft\_met}})$ is added to $P_{\text{MET}}$ (with different values event by event, by contrast with $\alpha_{\text{soft\_met}}$, which has a fixed value for a given pseudo-experiment) (this calculation is performed on the first two coordinates and $E_{\text{MET}}$ is recalculated from their modulus):

$$P_{\text{MET}}^{biased} = P_{\text{MET}} + \begin{pmatrix} Gauss(0, ET_{\text{soft}}) \\ Gauss(0, ET_{\text{soft}}) \end{pmatrix}$$

The corresponding modified PRImary features are then recomputed to new biased values: PRI_had_pt, PRI_leading_jet_pt, PRI_leading_jet_pt, PRI_met, and PRI_met_phi.

In addition,

$$PRI\_jet\_all\_pt^{\text{biased}} = \alpha_{\text{jes}} \times PRI\_jet\_all\_pt$$

If the number of jets is three or more, the impact of $\alpha_{\text{jes}}$ on missing ET cannot be calculated, given that detailed information on the additional jets (beyond two) is not available; this is a legitimate approximation as the total jet transverse momentum would be in most cases dominated by the first two leading.

DERived features are also impacted if they depend on these PRImary features (see Appendix C). Thus, for each of $\alpha_{\text{tes}}$, $\alpha_{\text{jes}}$ and $\alpha_{\text{soft\_met}}$, different features are impacted in a correlated way.

### D.3    Weight impacting bias implementation

$\alpha_{\text{bkg\_scale}}$, $\alpha_{\text{ttbar\_scale}}$ and $\alpha_{\text{diboson\_scale}}$ only impact the Weight of background events, more precisely:

- events with `DetailedLabel="ztautau"`:

$$\text{Weight}^{bias} = \alpha_{\text{bkg\_scale}} \times \text{Weight}$$

- events with `DetailedLabel="ttbar"`:

$$\text{Weight}^{bias} = \alpha_{\text{bkg\_scale}} \times \alpha_{\text{ttbar\_scale}} \times \text{Weight}$$

- events with `DetailedLabel="diboson"`:

$$\text{Weight}^{bias} = \alpha_{\text{bkg\_scale}} \times \alpha_{\text{diboson\_scale}} \times \text{Weight}$$

So $\alpha_{\text{bkg\_scale}}$ only affects the overall level of the background but leaves the background distributions unchanged. $\alpha_{\text{ttbar\_scale}}$ and $\alpha_{\text{diboson\_scale}}$ impacts only the proportion of the smaller backgrounds (see Table 1), thus distorting the overall background distribution.

### D.4    Event selection

Hadronic tau (and also the jets) can only be identified in the detector above a certain transverse momentum threshold ("low threshold" in the following) so that the raw dataset PRI_had_pt, PRI_jet_leading_pt PRI_jet_subleading_pt have clear thresholds. When applying $\alpha_{\text{tes}}$ and $\alpha_{\text{jes}}$, these thresholds move so that if nothing else is done, the threshold position would be an obvious giveaway of the value of $\alpha_{\text{tes}}$ and $\alpha_{\text{jes}}$.

To alleviate this, "high thresholds" (see Table 3) have been defined, which should systematically be applied after the calculation of the biased PRImary parameters, so that the thresholds to be observed on PRI_had_pt, PRI_jet_leading_pt PRI_jet_subleading_pt are independent of $\alpha_{\text{tes}}$ and $\alpha_{\text{jes}}$. The

| Variable | Low threshold (GeV) | High threshold (GeV) |
|---|---|---|
| $P_{\mathrm{had}}^{\mathrm{T}}$ | $\simeq 23$ | 26 |
| $P_{\mathrm{leading\ jet}}^{\mathrm{T}}$ and $P_{\mathrm{subleading\ jet}}^{\mathrm{T}}$ | $\simeq 23$ | 26 |

Table 3: Low and high threshold of hadronic tau and jet transverse momentum.

ranges in Table 2 are such that the thresholds should also be applied when no systematics bias is used[8].

---

[8]In practice, function `systematics` in `https://github.com/FAIR-Universe/FAIR_Universe_dataset/blob/main/hep_challenge/systematics.py` should always be used, even in the no systematics case.

