# OpenReview forum: "FAIR Universe HiggsML Uncertainty Dataset and Competition"
_NeurIPS.cc/2025/Datasets_and_Benchmarks_Track — NeurIPS 2025 Datasets and Benchmarks Track poster_

### Official Review · Reviewer_w3VM · 2025-06-29

**Rating:** 4
**Confidence:** 3

**Summary:**

The paper presents a report on a past NeurIPS competition: FAIR Universe HiggsML Uncertainty Dataset and Competition.
The authors release the dataset (and associated codebase) used for the competition for public use, and summarize the dataset creation process, the competition task and metrics, and the winners' solutions.

**Dataset Code Accessibility:**

Yes

**Ethical Considerations:**

No, there are no or only very minor ethics concerns

**Final Justification:**

Thank you for the response. Given the promises to fix the raised issues with the manuscript and the additional context around the usefulness of the provided dataset I have decided to raise my score from a borderline reject to a borderline accept, in alignment with the remaining reviewers.

**Limitations Weaknesses:**

This is a hard paper to evaluate as it is not a typical paper proposing a novel dataset and/or benchmark. It reads more like a historical report on a past competition. However, I find the paper to be generally poorly written and lacking a unifying story and good flow:
- For a paper proposing a new dataset I would expect there to be some motivation of why the task is important or relevant for future research. In general, I find that the paper fails to tell a coherent story and present a clear motivation for why this dataset is important, how it fits in uncertainty estimation research and how the authors expect it to be used in the future.
- More importantly, there is a lack of detail and clarity on the task itself. I think it would be important for the paper to somewhat stand on its own as a reference for the use of the dataset, how it was generated, what are the nuisance parameters, what are the main challenges. But the main body of the paper is very hard to follow and sparse in details.
- As an example of the previous point, there is insufficient detail on how the baseline from the starter kit works. What is the goal of the event selection and how is that achieved by training a classifier? Why are events split by score bins? What are the calibration curves? What is $S$ and $B$ (they are never properly defined)?
- The paper also lacks any sort of meta-analysis of the winner solutions. These are described in inconsistent levels of detail. Many of them are plagued by undefined symbols in the equations

**Strengths Contributions:**

The paper provides interesting insights into a past NeurIPS competitions and it's winning solutions.
It also provides a dataset and associated software that could be valuable for future research.

---

> ### Author Rebuttal · Authors · 2025-07-29
>
> We thank the referee on his/her assessment.
>
> On  “It reads more like a historical report on a past competition” : This statement of the referee is important. We were told that, as a selected competition in NeurIPS 2024, we had to submit to NeurIPS 2025 Dataset and Benchmark track, but with no specific guidance. We have chosen to deal both with the dataset (and biasing script), and the winning solutions in the limited space. For this reason, details on the dataset and biasing script are in appendices, and the two winning solutions are described in detail in separate papers, as referenced (btw the HEPHY one has just been accepted by Phys. Rev. D).
>  We can (and will) certainly improve the clarity, with the help of the critical reading by a Computer Science colleague, and address the specific points raised by the referee but only within the space constraint.
>
>  On the motivation : The dataset generation uses a specialised open-source particle physics Monte-Carlo simulator. No such open dataset exists in the field, by at least one order of magnitude. The size allows a precise evaluation of the uncertainty. The biasing script is completely original.
>
>  It is expected that new techniques (or previously published techniques) will be evaluated with the competition metric. (In fact, we see just now that it has been used in the talk Higgs Signal Strength Estimation with a Dual-Branch GNN under Systematic Uncertainties in early July 2025 EPS HEP Marseilles conference, a major physics conference).
>
> But new use of this large dataset and software can be foreseen, we can mention two:
>    * techniques to be evaluated with the same metric, but coupled with a data-frugality and/or compute frugality metric
>    * as indicated in the text the framework for this challenge was “known unknowns” :almost by definition, all the biases implemented in the public github are known unknowns. But it is easy to introduce new biases and check what happens if they are ignored. For example, TES simulates a constant scaling between the true and measured hadron Tau energy, but one can imagine a non linear effect, or a TES which would depend on the position in the detector (both effects happen in real life).  Another example: in addition to scaling of the background contributions, one could introduce distortions of these backgrounds.
>
> We’ll mention this in the text.
>
> The dataset has already more than 50 downloads. It is also well suited for teaching and training, we are aware so far of two very recent courses/summer school using it.
>
>
> On the baseline : the goal of using a classifier is to build a set of summary statistics (the score histogram bins) with improved signal (S) over noise (B) (with respect to the original sample), from which a binned likelihood fit can extract the signal strength mu. This is a “classical” technique in the field of High Energy Physics.
>
> We will clarify it and add more references
>
> On “Many of them are plagued by undefined symbols in the equations” : Apologies for these, this will be fixed and notations harmonised.

---

### Official Review · Reviewer_rtN5 · 2025-06-30

**Rating:** 4
**Confidence:** 2

**Summary:**

The submission introduces a large-scale dataset and challenge for estimating Higgs boson signal strength while quantifying both aleatoric and epistemic uncertainties. Participants develop models to provide 68.27% confidence intervals on the signal strength under systematic biases. The paper describes the dataset, evaluation metrics, baseline methods, and leading approaches including simulation-based inference, contrastive normalizing flows, and hybrid classifiers with uncertainty quantification. This work bridges physics and ML communities to advance methods for uncertainty-aware inference on large scientific datasets.

**Dataset Code Accessibility:**

Yes

**Ethical Considerations:**

No, there are no or only very minor ethics concerns

**Final Justification:**

Thank you for the response. My concerns are addressed. Therefore, I will maintain my original score.

**Limitations Weaknesses:**

1. In Figure 3, the IBRAHIME method generally performs well overall. However, as mu increases, its performance shows significant fluctuations, especially around mu = 3.0, where the performance degrades notably. The authors could provide more explanation or analysis regarding this behavior.

2. The dataset only covers "known unknowns", the biases are parameterized and provided, which may limit generalization to real-world scenarios with unanticipated sources of uncertainty. The author can explain more about future work.

**Strengths Contributions:**

1. The paper presents a significant contribution by releasing a large, realistic dataset with systematic biases, enabling benchmarking of uncertainty-aware ML methods in high-energy physics.
2. It bridges physics and ML communities, setting a new standard for comparing techniques under controlled conditions.
3. The methodology and dataset are well-documented, with clear descriptions, code, and scripts provided.

---

> ### Author Rebuttal · Authors · 2025-07-29
>
> We thank the referee on his/her assessment.
>
> On Weakness 1:
> Fig 3a fluctuation (essentially down) of the coverage for Ibrahim between 2.8 and 3 was understood after the end of the competition to be essentially an implementation bug, in the handling of the maximum truth value of mu being 3. It has disappeared after correction.  (For transparency, we’ve chosen to show the performances of the algorithm as submitted to the competition).
>
> There are much more details about Ibrahim's (and the two other) methods which could not fit in the limited space of the paper, but the two winners’ methods are detailed in separate referenced papers.
>
> On Weakness 2:
> The primary reason to limit the original challenge and, by extension, the benchmark to known unknowns is to maintain transparency and reproducibility.
> Almost by definition, all the biases implemented in the public github are known unknowns.
> But one value of this dataset and bias script is that it is easy to introduce new biases and check what happens if they are ignored.
>    * For example, TES simulates a constant scaling between the true and measure hadron Tau energy, but one can imagine non linear effect, or a TES which would depend of the position in the detector (both effects are real life).
>    * Another example: in addition to scaling of the background contributions, one could introduce distortions of these backgrounds.
> We’ll mention  this in the text.

---

> > ### Comment · Reviewer_rtN5 · 2025-08-02
> > **Response for the rebuttal**
> >
> > Thank you for the response. My concerns are addressed. Therefore, I will maintain my original score.

---

### Official Review · Reviewer_FFRj · 2025-07-01

**Rating:** 5
**Confidence:** 2

**Summary:**

This submission addresses the FAIR Universe - HiggsML Uncertainty Challenge, a dataset and competition focused on advancing machine learning methods for quantifying systematic uncertainties in particle physics measurements. The challenge asks participants to infer the signal strength $\mu$ (or relative frequency) of Higgs boson events and to provide a confidence interval for this variable. This confidence interval is intended to account for systematic biases that may be represented in the measurements contained in data, for example through flawed measurements of the features of an experiment. The dataset comprises 220 million simulated collision events, with 28 features per event. A crucial aspect of the challenge / benchmark is a biasing script that is provided to practicioners, allowing them to generate pseudo-experiments with varying values for signal strengths $\mu$ and various nuisance parameters that introduce systematic biases. The dataset is publicly available on Zenodo, and associated code is provided on GitHub. The paper describes the competition setup, metrics, and outlines the approaches of the top-performing teams.

**Dataset Code Accessibility:**

Yes

**Dataset Code Comments:**

- The dataset is publicly available on the Zenodo platform. It is provided in tabular format.
- Relevant code is available on GitHub.
- The code used to generate the dataset itself is publicly available.
- The winning teams have provided their codebases publicly.

**Ethical Comments:**

The authors state they could not think of possible misuse of the dataset, and neither can I. Existing assets look properly credited and new assets are well-documented.

**Ethical Considerations:**

No, there are no or only very minor ethics concerns

**Final Justification:**

The authors have addressed the questions I asked. My score already recommended acceptance and I decide to maintain this score.

**Limitations Weaknesses:**

My main comments on weaknesses revolve around clarity and presentation. I believe the presentation of the paper can make it hard at times for a reader not familiar with high-energy physics to quickly grasp the core issue and task of the challenge.

Clarity: The paper frequently uses the phrasing "systematic (epistemic) uncertainties", but the discussion around this concept could benefit a further explicit clarification for a broader ML audience. Given that participants have access to a biasing script that allows them to perform (in theory) infintely many forward simulations of the data generating process, the "epistemic" nature of the involved uncertainty goes against my intuition. Could one not phrase this as noise that is due to unobserved variables?

Scope of uncertainties: This point is related to the previous, and the paper also states this explicitly: "The main limitation of the setup is that biases can be exactly parameterised: we are in the 'known unknowns' regime". I would be interested to see a more explicit elaboration of what the authors would envision as a potential "unknowns unknowns" setting. For example, could there be entire unobserved biasing mechanisms that result in previously never observed patterns of features? This would be a highly interesting extension to what I more easily view as epistemic uncertainties.

Competition-specific hacks: The paper acknowledges an artificial "hack" for the competition: "The CI width is seen falling at large values of $\mu$: this is due to the clipping the Confidence Interval to a maximum value of 3, .... such clipping would be meaningless in the context of a real physics measurement where μ is truly unknown". Could this artifact be solved easily and what is the influence of this hack being possible on the resulting algorithms?

Clarity: Some aspects of the competition setting took me longer to grasp than necessary. For example, it took me some effort to understand what role the biasing script is inteded to play in the participants space of solution. This could perhaps be highlighted more in the "Tasks and application scenarios" section. Initially, this script sounded to me more like an addendum provided "for completeness" rather than as a core component. I was also unsure whether the script was inteded to be used for example during online inference in the evaluation algorithm.

**Strengths Contributions:**

The work represents a tremendous effort and makes significant contributions to the intersection of machine learning and high-energy physics, particularly in the critical area of uncertainty quantification.

Novelty: The presented challenge is highlighted as the "first challenge and dataset that requires participants to handle systematic uncertainty". I believe this is a crucial direction that will play a big role at the intersection of machine learning and scientific discovery, yet is surprisingly often overlooked. This challenge explicitly requires the computation of confidence intervals, incorporating both aleatoric and epistemic uncertainties.

Scale: The dataset, consisting of 220 million instances and 28 features, is substantially larger than many previous benchmarks. This is crucial in my view and allows for the development and benchmarking of models at scale. Recent trends in the field show that this is the setting where many major developments are achieved and the scale of this benchmark is thus an important factor in its impact to the field.

Biasing Script: I particularly like the inclusion of a script to parameterize and apply systematic biases. This allows participants to explicitly generate data with known systematic biases, which is crucial and rare for datasets at this scale. The script was actively utilized by the winning solutions.

Public Accessibility: The dataset and relevant code are publicly available.

---

> ### Author Rebuttal · Authors · 2025-07-29
>
> We thank the referee on his/her assessment.
>
> On: “allow them infinitely many forward simulations of the data generating process” We guess that the referee means that, after all, this can be seen "just" as a regression problem of 7 parameters (mu and the six Nuisance Parameters (NP)). This is an important remark, but the fact that we care only about mu precision and coverage, and not at all about the NP's is what makes it "epistemic". We'll clarify this.
>
> On:”I would be interested to see a more explicit elaboration of what the authors would envision as a potential "unknowns unknowns" setting”
> Almost by definition, all the biases implemented in the public github are known unknowns.
> But one value of this dataset and bias script is that it is easy to introduce new biases and check what happens if they are ignored.
>    * For example, TES simulates a constant scaling between the true and measured hadron Tau energy, but one can imagine a non linear effect, or a TES which would depend of the position in the detector (both effects happen in real life).
>    * Another example: instead of scaling of the different background fractions, one could introduce distortions of these backgrounds.
> We’ll add something to this matter in the text
>
> On: “Could this artifact be solved easily and what is the influence of this hack being possible on the resulting algorithms?”
> Fig 3b shows that the influence of this hack for the best two methods is limited to above 2.3, so it does not have any fundamental role. Increasing the maximum value of the true mu to larger values (3 was an arbitrary choice) would limit even further the impact of the hack. Another possibility would be to give to true mu a non bounded distribution, but it could also be exploited by participants in a more sophisticated, less visible way.
> By the way one can see a similar effect on the lower value sides, which could be limited further by reducing the lower limit of mu from 0.2 to 0 (0 being the physical limit).
>
> On clarity: we will work on improving it with the help of the critical reading by a Computer Science colleague.

---

> > ### Comment · Reviewer_FFRj · 2025-08-04
> >
> > Thank you for the rebuttal and the work. I find that my questions / comments have been addressed and I will keep my score.

---

### Official Review · Reviewer_Ap8j · 2025-07-03

**Rating:** 5
**Confidence:** 3

**Summary:**

This paper introduces the FAIR Universe HiggsML Uncertainty Challenge, a novel benchmark dataset and challenge focused on quantifying systematic uncertainties in high-energy physics (HEP) measurements. The authors present a large-scale tabular dataset (220M simulated proton-proton collision events with 28 features) that models Higgs boson decays to tau leptons while incorporating six parameterized systematic biases. The challenge required participants to estimate confidence intervals for the Higgs signal strength while accounting for these epistemic uncertainties, with evaluation metrics balancing interval width and coverage accuracy. Three winning approaches are highlighted, including Contrastive Normalizing Flows and simulation-based inference methods.

**Dataset Code Accessibility:**

Yes

**Ethical Considerations:**

No, there are no or only very minor ethics concerns

**Final Justification:**

Given the authors' response and the reviews from other reviewers, I maintain my original score, leaning toward a positive rating.

**Limitations Weaknesses:**

(1) While datasets/code are available (Zenodo[65], GitHub[74]), key implementation details for the winning methods are omitted due to space constraints (e.g., hyperparameters in CNFs[79], architecture specifics in GOLLUM[76]). This may limit reproducibility of the top results. ​​

(2) While the paper makes commendable efforts to bridge HEP and ML communities, its presentation could be more approachable for readers without particle physics expertise, as NeurIPS is a machine learning conference.

**Strengths Contributions:**

*  ​**Impactful Dataset Design​​:** The dataset represents a significant advancement over previous benchmarks (e.g., HiggsML) by incorporating realistic systematic uncertainties at scale (220M events vs. <1M in prior work). The inclusion of both primary particle attributes and derived features enables research on both low-level and high-level representations. ​​

* **Novel Evaluation Framework​​:** The asymmetric scoring metric (Equation 2) innovatively addresses High-Energy Physics (HEP) needs by penalizing under-coverage more severely than over-coverage, while the Neyman construction (Figure 5 right) provides statistically rigorous interval estimation.

* **​​Interdisciplinary Bridge​​:** The work successfully connects ML uncertainty quantification techniques with HEP requirements, as evidenced by the diverse winning solutions (Section 5). The public release on Zenodo platform with Croissant metadata and biasing scripts enables long-term benchmarking.

---

> ### Author Rebuttal · Authors · 2025-07-29
>
> We thank the referee on his/her assessment.
>
> On Weakness 1: as a selected NeurIPS 2024 competition, we were asked to submit to NeurIPS 2025 Dataset and Benchmark track with no precise guidelines on how to do so. We’ve chosen to deal with both the dataset (and accompanying software) and the winning solutions, hence the lack of details. However, the top two submissions are fully documented in separate papers as referenced, HEPHY’s has already been accepted by Phys. Rev. D.
>
> On Weakness 2: we will clarify further for a non physicist reader with the help of the critical reading by a Computer Science colleague.

---

### Decision · Program_Chairs · 2025-09-18

**Decision:**

Accept (poster)

**Comment:**

The paper was originally from a NeurIPS 2024 competition called The Fair Universe HiggsML Uncertainty Challenge. The paper details the dataset and discusses the competition results. All reviewers were positive about this work and voted for acceptance. They generally found the evaluation framework novel and the dataset playing an important role for briding physics and ML. On the negative side, some reviewers complained about the clarity issue and found the paper hard to read as some terms are not easy to understand for the general ML audiences. After carefully reading the reviews, rebuttal, and paper, the AC agrees with the reviewers that the paper has potential for advancing ML for particle physics.